# Transplanting FVIII/ET3-secreting cells in fetal sheep increases FVIII levels long-term without inducing immunity or toxicity

Martin Rodriguez [1], Brady Trevisan[1], Ritu M. Ramamurthy [1], Sunil K. George[1], Jonathan Diaz [1], Jordan Alexander[2], Diane Meares[3], Denise J. Schwahn [4], David R. Quilici[5], Jorge Figueroa[6], Michael Gautreaux [7], Andrew Farland[3], Anthony Atala [1], Christopher B. Doering [2], H. Trent Spencer [2], Christopher D. Porada [1,8] & Graça Almeida-Porada [1,8] ✉

Hemophilia A is the most common X-linked bleeding disorder affecting more than half-a-million individuals worldwide. Persons with severe hemophilia A have coagulation FVIII levels <1% and experience spontaneous debilitating and life-threatening bleeds. Advances in hemophilia A therapeutics have significantly improved health outcomes, but development of FVIII inhibitory antibodies and breakthrough bleeds during therapy significantly increase patient morbidity and mortality. Here we use sheep fetuses at the human equivalent of 16–18 gestational weeks, and we show that prenatal transplantation of human placental cells ($10^7$–$10^8$/kg) bioengineered to produce an optimized FVIII protein, results in considerable elevation in plasma FVIII levels that persists for >3 years post-treatment. Cells engraft in major organs, and none of the recipients mount immune responses to either the cells or the FVIII they produce. Thus, these studies attest to the feasibility, immunologic advantage, and safety of treating hemophilia A prior to birth.

Hemophilia A (HA) the most common X-linked bleeding disorder, occurs in 24.6 per 100,000 live male births and affects over half-a-million individuals worldwide[1]. Persons with severe HA (PHA) have <1% of the normal plasma levels of FVIII activity, and endure life-threatening spontaneous bleeding, including connective tissue/muscle hematomas, internal bleeding, and hemarthroses leading to chronic debilitating arthropathies[2]. Despite the latest advancements in current therapies, which have substantially advanced the standard of care[3], the burden of disease in PHA continues to be high, as joint disease, development of inhibitory antibodies in 25–40% of patients, within the first 50 exposure days to FVIII, and prevention of breakthrough and/or life-threatening bleeding continues to be a challenge[4,5].

A recent study demonstrated that for those born with hemophilia, the chances of living a life of normal duration and quality will be reduced by 33% in high income countries, and by 93% in low-income countries[1]. These sobering statistics demonstrate that there are still urgent unmet clinical needs requiring the development of novel therapeutic approaches to increase survival and quality of life in these patients.

Treatment of genetic disorders during the prenatal period using in utero transplantation (IUTx) has safely been performed for decades

[1]Wake Forest Institute for Regenerative Medicine, Fetal Research and Therapy Program, Wake Forest School of Medicine (WFSOM), Winston Salem, NC, USA. [2]Aflac Cancer and Blood Disorders Center, Children's Healthcare of Atlanta and Department of Pediatrics, Emory University, Atlanta, GA, USA. [3]Special Hematology Laboratory, Wake Forest School of Medicine, Winston Salem, NC, USA. [4]Preclinical Development, Wave Life Sciences, Lexington, MA, USA. [5]The Mick Hitchcock Ph.D. Nevada Proteomics Center, University of Nevada Reno, Reno, NV, USA. [6]Center for Research in Obstetrics and Gynecology, WFSOM, Winston Salem, NC, USA. [7]HLA/Immunogenetics and Immunodiagnostics Laboratories, Winston Salem, NC, USA. [8]These authors contributed equally: Christopher D. Porada, Graça Almeida-Porada. ✉e-mail: galmeida@wakehealth.edu

in humans under compassionate use[6]. Recently, clinical trials recruiting subjects to establish the safety and feasibility of using IUTx to treat fetuses with α-thalassemia (NCT02986698), osteogenesis imperfecta (NCT03706482), or lysosomal storage diseases (NCT04532047) were initiated. These trials attest to the much-needed change in perspective of how patients with monogenic disorders can be treated, and have further established the safety of this type of procedure(s) for both pregnant women and the affected subject[7].

Approximately 70% of persons with HA have a family history of the disease, allowing for prenatal diagnosis and intervention. The development of prenatal therapies able to provide FVIII levels that are curative, or sufficient to convert a severe, life-threatening bleeding disorder to a mild phenotype, while concomitantly inducing immune tolerance to the FVIII product, and thereby eliminating the risk of FVIII inhibitor formation, would have a major beneficial impact on the life expectancy and quality-of-life of HA patients, and it would transform the standard of care in HA treatment[8,9].

While in utero gene therapy has shown great promise in preclinical studies[10], transduction of cells in vitro allows safeguards in production that are not possible with direct vector injection and eliminates the risk of inadvertently transducing undesired tissues/cells, e.g., those of the germline. Furthermore, IUTx using cells that proliferate physiologically in vivo and secrete clinically therapeutic levels of FVIII is a promising and safe approach for providing long-term/permanent correction of HA[6]. We have recently reported that human placental cells (PLC) are an excellent cell platform to produce and secrete FVIII in vivo when transduced with a lentivector encoding a myeloid codon-optimized, bioengineered FVIII (PLC-mcoET3)[11]. Here, using wild-type sheep as a large animal model of IUTx, we demonstrate that administration of PLC-mcoET3 at the equivalent of 16–18 gestation weeks (g.w.) in humans, resulted in elevated plasma FVIII levels that exceeded those of control non-IUTx animals by >48.4 ± 12.3%, for >3 years after birth, despite the rapid and considerable increase in weight, with no evidence of therapy-related toxicity, nor the development of anti-FVIII/ET3 IgGs, FVIII inhibitors, or ET3-specific $T_{h1}$ or $T_{h2}$ cells. In addition, we showed that IUTx-treated recipients did not develop immunity or anti-HLA antibodies to the transplanted human PLC but maintained robust reactivity to foreign antigens. RNA and DNA analysis proved engraftment and continued expression of the vector-encoded mcoET3 in all major organs. This therapy also resulted in the absence of bleeding of a HA sheep, despite a challenging birthing. These studies thus attest to the feasibility, immunologic advantage, and safety of treating HA during the prenatal period.

## Results

### Prenatal transplantation of human PLC-mcoET3 results in increased levels of ET3 protein in circulation for at least 3 years after treatment

A total of 25 animals were transplanted in utero (IUTx) with human placenta-derived mesenchymal cells (PLC) transduced with a lentiviral vector encoding mcoET3, a myeloid-codon-optimized, bioengineered FVIII transgene[12], (PLC-mcoET3) at 59–65 gestational days (g.d.), which corresponds to ~16–18 g.w. in humans, at estimated doses of $10^7–10^8$ cells/kg, as detailed in the study design and Supplementary Table 1. PLC-mcoET3 master cell bank used in this study was previously described regarding isolation, culture, phenotypical characterization, prior and after transduction[11]. Animals were followed for one ($n = 13$) to three ($n = 8$) years after IUTx. Evaluation of plasma FVIII activity at intervals throughout the duration of the study, expressed as % increase over non-transplanted control group, showed that, throughout the first year, treated animals exhibited an overall 54.3 ± 7.5% increase ($n = 13$) in plasma FVIII activity (Fig. 1a); of note, normal hemostasis requires at least 25% of factor VIII activity. Importantly, the elevated FVIII activity was maintained despite the exponential growth of the animals, which weighed ~0.1 kg at the time of transplant, 3.7 ± 0.34 kg

at birth, and 62 ± 4.1 kg at 1 year of age (Fig. 1b). From 12 to 24 months after birth, the mean increase in FVIII activity continued to be at levels of 47.4 ± 16.8% ($n = 10$), ranging from 14%–191% in eight out of nine animals, with one failing to reach 5% FVIII increase (Fig. 1c). Evaluation of eight animals during the 3rd year after IUTx demonstrated an overall mean FVIII activity increase of 48.4 ± 12.3%, with two animals showing FVIII activity between 5 and 20% (Fig. 1d). To determine if the % of FVIII was declining significantly with time after transplant, analysis was performed evaluating differences between mean levels of increased FVIII activity during the first and the third year after transplant, and results are shown in Supplementary Fig. 1. All but one recipient maintained (or did not change significantly) levels of FVIII activity despite the steady increase in weight, which at 36 months reached 124 kg and 95 kg for males and females, respectively (Fig. 1b).

To confirm the presence of ET3 protein in the plasma of treated animals, liquid chromatography-tandem mass spectrometry (LC-MS) was performed. Data independent analysis was used to provide the relative abundance of ET3 within each animal's plasma and demonstrated that the increase in FVIII activity was consistent with the presence of ET3 protein in circulation. Figure 1e shows the exclusive intensity which reflects the summarized intensity value of the peptides only associated with this protein.

### Sustained presence of ET3 protein in circulation does not induce anti-ET3 IgM and IgG antibodies

We have previously shown that administration of ET3 or hFVIII protein to wild-type juvenile sheep induces a strong immune anti-ET3/FVIII IgG response starting at week 2 post-infusion and leads to the development of inhibitory antibodies[13]. To investigate if circulating ET3 protein provided by the transduced cells induced anti-ET3 antibodies in IUTx animals, ET3-specific IgM and IgG ELISAs were performed in plasma at multiple time points after birth. As shown in Fig. 2a, b, during the first year post-IUTx, none of the treated animals developed anti-ET3 IgM or IgG antibodies. Similarly, during the second year (Fig. 2c), all but 1 recipient continued to be devoid of anti-ET3 IgG antibodies. The one animal (17010) that exhibited a low titer (1:20) anti-ET3 IgG antibody at 18 months post-transplant, was closely followed with subsequent anti-ET3 IgG antibody testing, and data showed that the anti-ET3 IgG antibody was either transient or an assay artifact. Detailed results of anti-ET3 IgG antibody testing performed from 4 to 24 months in this animal are shown in Fig. 2d.

### Absence of ET3-specific memory $T_{h1}$ and $T_{h2}$ lymphocytes in IUTx recipients

To determine whether treated animals had developed ET3-specific memory T cells, IFN-γ and IL-4 ELISpot assays were performed to determine the presence, secretion intensity, and frequencies of reactive ET3-specific memory T cells. None of the IUTx recipients harbored ET3-specific $T_{h1}$ or $T_{h2}$ lymphocytes (Fig. 3a, b respectively), while their lymphocytes maintained the ability to react and secrete IFN-γ and IL-4 when stimulated with Phytohemagglutinin-L (PHA-L), thereby demonstrating that their lymphocytes were functional, able to be activated, and capable of secreting the assayed cytokines.

### Human PLC-mcoET3 are not immunogenic and do not induce donor-specific cellular immune responses

To test whether PLC-mcoET3 were immunogenic and could sensitize fetal recipients to the administered therapy, one-way Mixed Lymphocyte Reactions (MLR) were performed by co-culturing recipients' peripheral blood mononuclear cells (PBMC) with (1) same-donor PLC-mcoET3; (2) same-donor non-transduced PLC; (3) xenogeneic (third-party human) PBMC, or (4) individual self-PBMC to establish the baseline stimulation index. None of the IUTx recipients had significant change in PBMC proliferation when co-cultured with either transduced PLC or non-transduced PLC and compared to co-culture with

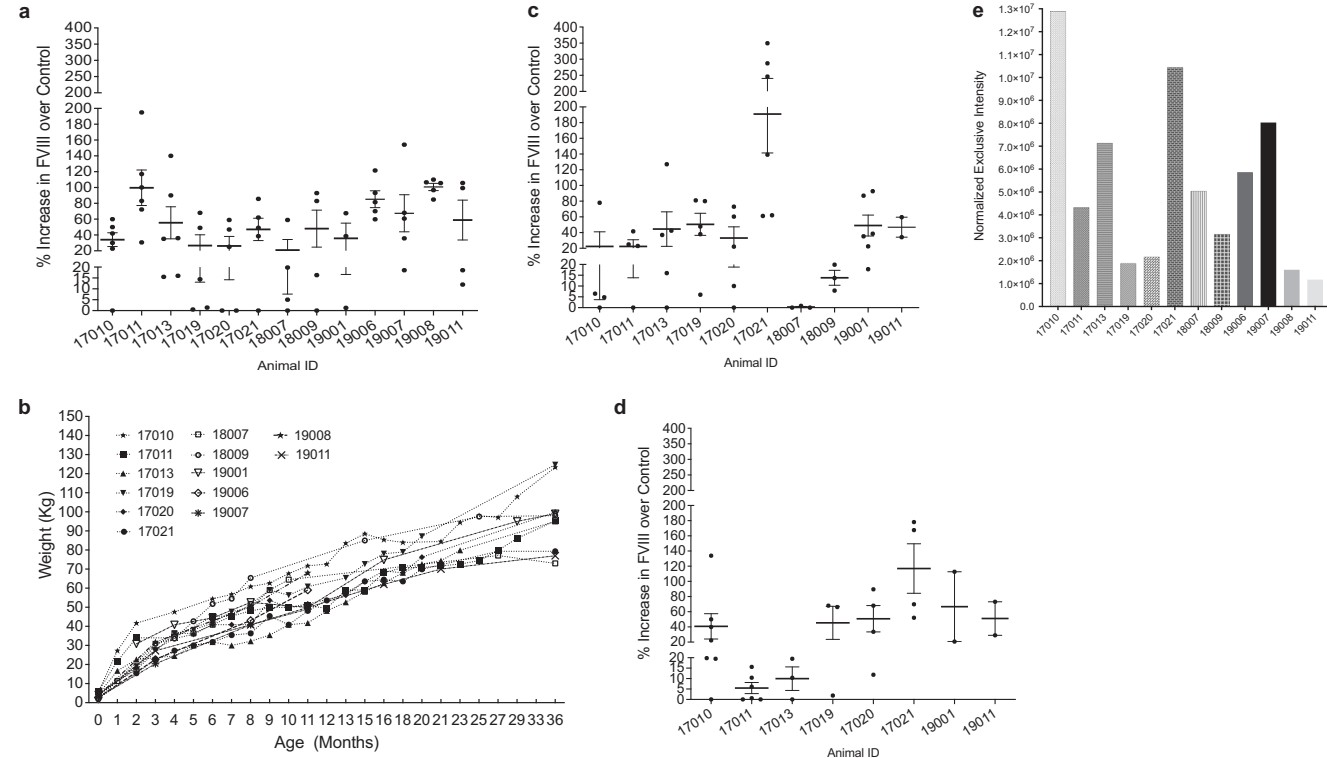

**Fig. 1 | Prenatal transplantation of human PLC-mcoET3 results in increased FVIII activity levels in circulation for at least 3 years after treatment.**
**a** Evaluation of FVIII levels by FVIII activity assays during Year 1 after birth demonstrated that all treated animals exhibited an increase in plasma FVIII activity that would be able to provide normal hemostasis ($n = 13$ animals; each animal was tested at $n = 3$ to $n = 6$ different time points, as indicated by dots/animal); **b** Weight-curve demonstrated that the increase in FVIII activity was maintained despite the exponential growth of the animals ($n = 13$ animals); **c** Year 2 after birth, the mean increase in FVIII activity continued to be elevated in nine animals and one failed to reach 5% FVIII increase ($n = 10$ animals; each animal was tested at $n = 2$ to $n = 6$ different time points, as indicated by dots/animal); **d** Year 3 year after birth, six out

of eight animals maintained a considerable increase in FVIII activity, while the FVIII activity in the other two was between 5 and 20% ($n = 8$ animals; each animal was tested at $n = 2$ to $n = 6$ different time points, as indicated by dots/animal); **e** Confirmation of the presence of ET3 protein in the plasma of treated animals by LC-MS, demonstrating that the increase in FVIII activity detected by FVIII activity assays was due to the presence of ET3 protein in circulation. LC-MS was used as a targeted approach and the normalized exclusive intensity reflects the summarized intensity value of the peptides only associated with this protein and thus providing the relative abundance of ET3 within each sheep. Bold bars depict the Mean ± SEM. Source data are provided as a Source Data file.

individual self-PBMC (Fig. 3c). To investigate whether IUTx recipient's PBMC had become anergic, IUTx recipient's PBMC were stimulated with third-party human PBMC and exhibited significantly increased cell proliferation when compared to the respective control, demonstrating that tested PBMC were functional and maintained the ability to recognize and respond to foreign antigens (Fig. 3c).

## Human PLC-mcoET3 do not induce anti-HLA Class I and Class II antibodies after IUTx

To explore whether PLC-mcoET3 administration elicited a humoral immune response that led to the formation of HLA Class I and/or Class II IgG antibodies, sequencing was first used to define the PLC HLA Type used in this study: A*02,24; B*18,51; C*02,07; DRB1*04,16; DRB4*01; DRB5*02; DQB1*03(DQ7),05; DQA1*01,03; DPA1*01; DPB1*04. Next, solid phase pooled bead assays were performed using a Luminex 200 multiplex system, modified for detection of antibodies in sheep serum. Results of this screen showed that none of the IUTx recipients had been sensitized against the donor cells, and their panel reactive antibody status (PRA) was considered negative (Fig. 4a). Of note is that one animal (17021), was reactive to HLA-B45, but since the transplanted cells do not contain this HLA antigen, this was considered a non-specific antibody. Another animal (18007) also displayed multiple non-specific xenogeneic HLA antibodies. Therefore, we also investigated whether (control) animals that were never transplanted with human cells had naturally occurring xenogeneic antibodies, that were

cross-reactive with the human-HLA bead targets. Results showed that all controls had xenogeneic antibodies with a broad range of MFIs (Fig. 4a). Assessment of anti-HLA-II antibodies in the sera of transplanted animals and non-transplanted control animals showed that none of the animals exhibited a positive PRA, or at least a positive PRA specific to the transplanted cells (Fig. 4b).

## Prenatal transplantation of PLC-mcoET3 does not cause hepatic or hematological alterations

To evaluate whether hepatic or hematological alterations were induced as a result of IUTx, white blood cell counts (WBC), hematocrits, and the liver enzymes, aspartate aminotransferase (AST) and alanine aminotransferase (ALT) were analyzed at intervals after birth. All transplanted animals maintained normal ALT and AST levels (Fig. 4c) and had hematocrits (Fig. 4d) and WBC (Fig. 4e) within normal range, that did not differ significantly from intra-flock non-transplanted group.

## Evaluation of sites of human PLC-mcoET3 engraftment after IUTx

To determine the sites and relative percentage of PLC-mcoET3 engraftment in different organs of animals treated prenatally ($n = 4$), recipients were euthanized at birth, which corresponds to approximately 3 months post-transplant. Of particular significance, one of these animals had been diagnosed during gestation to have HA, using

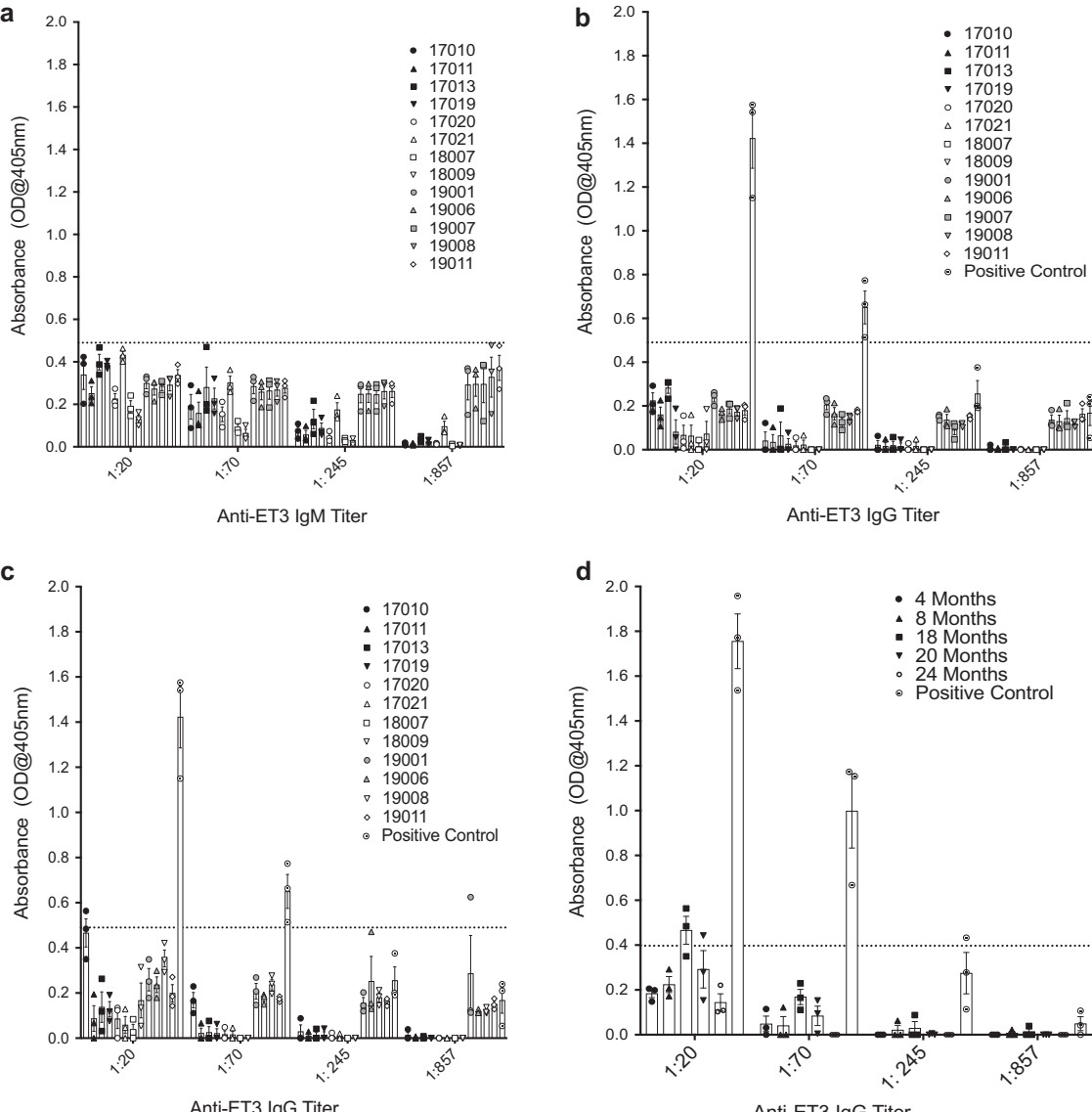

**Fig. 2 | Sustained presence of ET3 protein in circulation does not induce anti-ET3 IgM or IgG antibodies.** Anti-ET3 specific ELISAs were performed on serial dilutions of plasma of IUTx animals at multiple time points after birth (3–8 and 18 months) to investigate if circulating ET3 protein induced: **a** IgM ($n = 13$ animals) or **b** IgG antibodies in IUTx animals: each animal was tested at least twice, at (**b**) an earlier ($n = 13$ animals) and at (**c**) a late time point ($n = 12$ animals), and for each time point ELISAs were performed in triplicate. Positive antibody titers were defined as the dilution of plasma with an absorbance value > 2 SD above the mean OD from ELISAs performed with control sheep plasma (dotted line). During the first year post-IUTx, none of the treated animals developed anti-ET3 IgM or IgG antibodies, but during year 2, **c** a low-titer (1:20) anti-ET3 IgG antibody was detected in one animal (17010). This animal was closely followed with subsequent anti-ET3 IgG antibody testing, and data showed that (**d**) the anti-ET3 IgG antibody measured at different time points ($n = 5$) was either a transient non-inhibitor antibody, since FVIII levels remained high, or an assay artifact as subsequent samples from this animal were devoid of anti-ET3 IgG. Data are shown as Mean ± SEM. Source data are provided as a Source Data file.

DNA isolated from amniotic fluid cells and restriction fragment-length polymorphism (RFLP) PCR[14]. The diagnosis was also confirmed at birth using DNA from skin fibroblasts. The HA animal was stillborn as a result of labor and delivery complications, and despite the traumatic birthing process, at necropsy, the animal did not display evidence of the extensive abnormal bleeding, that has been reported in this line of HA animals[14,15].

RT-qPCR was performed, using mcoET3-specific primers on RNA isolated from liver, lung, spleen, and thymus of this HA animal and the other euthanized IUTx recipients. To create a standard curve for quantification, RT-qPCR was performed on RNA isolated from different percentages of PLC-mcoET3 mixed with sheep stromal cells. Results obtained from the different tissues were then extrapolated into

percentages using the standard curve. Percentages of PLC-mcoET3 engraftment in organs of treated animals are depicted in Fig. 5a, demonstrating that cells engrafted in all organs analyzed. The same tissue samples harvested from control non-treated animals showed no target amplification. qPCR using human Alu-specific primers in DNA extracted from the same tissues was performed to determine the overall presence of transduced and non-transduced human cells. A standard curve generated from DNA extracted from sheep tissues with increasing amounts of human DNA (0, 0.25, 0.5, 1, 5, and 10 ng) to a total 100 ng of DNA per sample was used to extrapolate the amount of human genomic DNA present in each animal and in each organ (Supplementary Fig. 2). All the animals and the tissues analyzed contained human DNA. While similar amounts of human DNA were found

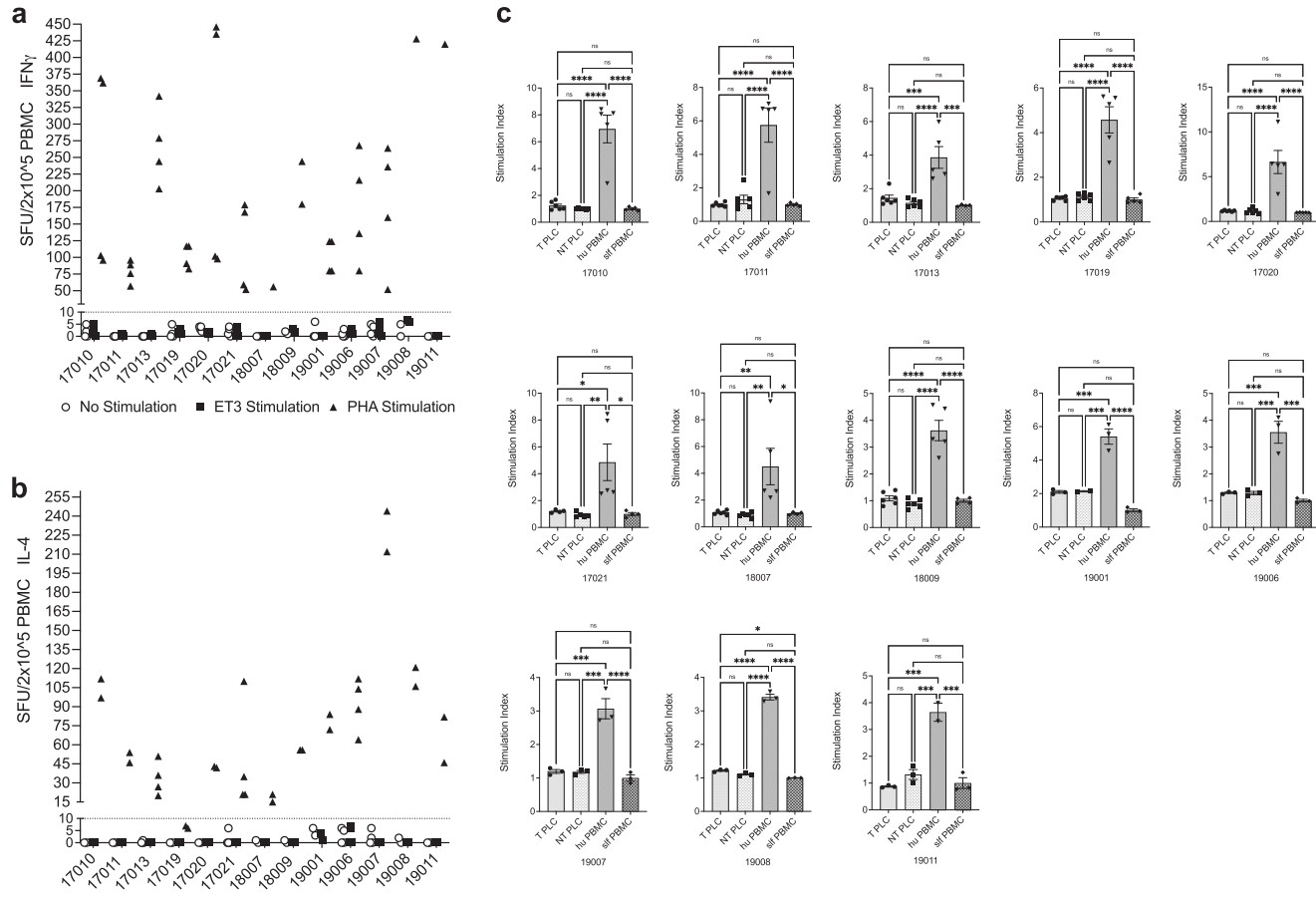

**Fig. 3 | Absence of cellular immune responses against ET3 and transduced (T) and (NT) nontransduced donor cells. a** IFN-γ and **b** IL-4 ELISpot assays were performed to determine the presence of reactive ET3-specific memory T cells. Assays were run in duplicate and from cells collected from each animal (*n* = 13 animals) on at least two different time points ranging from 5–20 months after birth. A positive response was defined by a Stimulation Index >2, and >10 SFU/2 × 10⁵ PBMC (dotted line). IUTx recipients (*n* = 13 animals) had no ET3-specific T_h1 or T_h2 lymphocytes, while their lymphocytes maintained the ability to react and secrete IFN-γ and IL-4 when stimulated with Phytohemagglutinin-L (PHA-L); **c** None of the IUTx animals' (*n* = 13 animals) lymphocytes proliferated in the presence of same-donor PLC-mcoET3 (T) or same-donor non-transduced PLC (NT), when tested by one-way MLRs, but maintained significant proliferative response to third party human PBMC. For each animal, MLR assays were performed in triplicate at two different time points. Data are shown as Mean ± SEM. One-way ANOVA followed by Tukey's multiple comparison test was used to determine significant differences. *p* ≤ 0.05 was considered significant. \**p* ≤ 0.05; \*\**p* ≤ 0.01; \*\*\**p* ≤ 0.001; \*\*\*\**p* ≤ 0.0001. Source data are provided as a Source Data file.

in the spleen and thymus, different animals differed in the amounts of human DNA present the liver and lung (Supplementary Fig. 2).

Since the liver is a major site of FVIII production, distribution and localization of transplanted cells in this organ were also examined by immunohistochemistry, using an antibody specific for the human nuclear protein, Ku80, and slides were quantified using image J (Fig. 5b). Representative images of Ku80 staining in liver can be seen in Fig. 5c–f. Supplementary Fig. 3 also depicts representative images of Ku80 staining in the thymus of these animals.

In addition, and to better visualize the location of PLC-mcoET3 within the liver tissue and any histomorphological alterations caused by PLC-mcoET3 at the site of engraftment, Ku80 staining was also performed using the chromogen 3,3-diaminobenzidine (DAB). The Ku80 positive dark nuclei of the human cells can readily be identified within the parenchyma of the different liver sections (Fig. 6c–f). There is no evidence of either PLC-mcoET3 clonal proliferation or of disruption of the liver cellular architecture at the locale where the cells lodged.

The production and presence of ET3 protein in liver was also determined in the IUTx HA animal (Fig. 5g, h). Since animals with this mutation lack factor VIII antigen (cross-reactive material negative) (Supplementary Fig. 4), detection of FVIII/ET3 in the IUTx HA animal results from the production of ET3/FVIII by the transplanted cells

present in the organ. Of note is the presence of ET3/FVIII-producing cells between the hepatocytes and interlaced with the hepatic sinusoids. Supplementary Fig. 5 depicts ET3/FVIII-positive cells located in the thymus of the HA animal.

### PLC-mcoET3 persist long-term in the engrafted tissues
FVIII activity in plasma of IUTx animals remained elevated throughout the 3rd year of analysis, suggesting that PLC-mcoET3 persist long term in the engrafted tissues. To verify that PLC-mcoET3 were present long-term in the different organs, six animals that were not HA carriers or male breeders, were euthanized at 2.3–5 years post IUTx (Supplementary Table 1). qPCR using human-Alu specific primers on DNA extracted from liver, lung, spleen, and thymus was performed as described above, and the results shown in Fig. 7a–e, demonstrate that human DNA was still present in most organs analyzed. The same tissue samples harvested from control non-treated animals showed no target amplification. To confirm that the human DNA detected corresponded to engrafted human cells and determine their location, tissue sections from livers of these animals were also examined by immunohistochemistry, using antibodies specific for the human nuclear protein, Ku80, and human albumin. Representative images of liver tissues from human, control non-transplanted sheep and IUTx animals can be seen

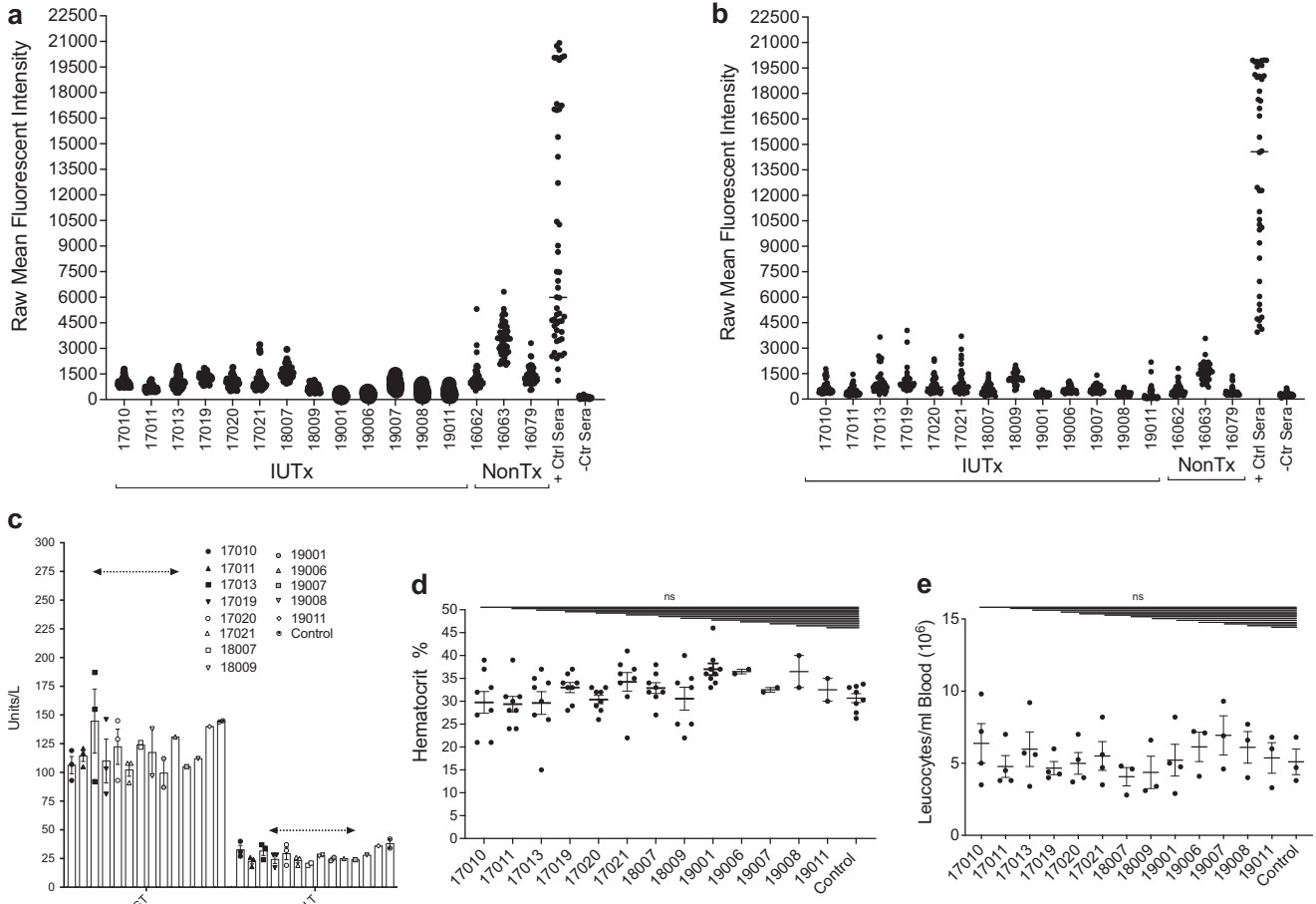

**Fig. 4 | Human PLC-mcoET3 do not induce anti-HLA Class I antibodies after IUTx, and administration of the product does not cause hepatic or hematological alterations.** Raw mean fluorescence intensities (MFI) results from solid phase pooled bead assays, modified for detection of (**a**) HLA-I and (**b**) HLA-II antibodies in IUTx sheep ($n = 13$ animals) sera are shown. Also included are the results from positive control beads, and non-treated control animals. For each animal, dots correspond to MFI of each bead in the panel. If MFI values in ≥2 beads were negative, their panel reactive antibody status (PRA) was considered negative. None the transplanted animals were found to have a positive PRA specific to the transplanted cells; **c** Alanine aminotransferase (ALT) and Aspartate aminotransferase (AST) ($n = 13$ animals); **d** hematocrit values ($n = 13$ animals); and **e** WBC ($n = 13$ animals) were analyzed at different times after birth, each represented by a dot. No statistically significant differences (ns) were found between IUTx treated animals and a pool of control animals ($p \leq 0.05$). Data are shown as mean ± SEM. One-way ANOVA followed by Dunnett's multiple comparisons tests were used to determine significant differences between each animal and the control animal. Dotted lines denote upper normal range of AST and ALT. Source data are provided as a Source Data file.

in Fig. 7f–k. The dark nuclei of human cells can be readily identified nearby liver sinusoids, while none of the hepatocytes in the slides analyzed are positive for human albumin.

To ascertain that mcoET3 was still being transcribed by the engrafted cells, RT-qPCR was performed on RNA isolated from liver, lung, thymus, and spleen using mcoET3-specific primers, as described above (Supplementary Fig. 6a, b–e). In addition, RNA isolated from reproductive tissues and from thoracic and mesenteric lymph nodes was also analyzed by RT-qPCR (Supplementary Fig. 6a, f–h).

### Histopathological evaluation of tissues from IUTx animals

To investigate whether engraftment of PLC-mcoET3 in the different tissues had caused untoward effects, H&E staining was performed on tissues sections of all animals euthanized and the slides sent to a certified animal pathologist. Results showed no evidence of any lentiviral-related or procedural toxicity in any of the tissues examined. In addition, there were no changes in tissue architecture, ectopic tissue formation, gross tumor development, or areas of atypical cells or foci of hyperplasia or neoplasia. Representative images of these tissue sections can be seen in Fig. 8.

### PLC-mcoET3 persist long term in the engrafted tissues and upregulate several tissue cell-specific transcripts

To investigate whether organ resident PLC-mcoET3 assumed identities consistent with specific cell types within the organs, RNA was isolated from the main tissues of the animals and transcriptome RNA-seq libraries were created, which were aligned to Homo sapiens libraries after eliminating any possible cross-reactivity with ovine transcripts. RNA was also isolated from PLC-mcoET3 prior to transplant, and transcriptome RNA-seq libraries created. Transcripts of the PLC-mcoET3 were compared with human transcripts known to be expressed in organ-specific cells, and with the human transcripts found in the same organs of the transplanted animals. The results can be found in Supplementary Tables 2–6, which depict human liver-, lung-, thymic- and ovary/testes-specific upregulated transcripts. These results show that, although several cell-specific transcripts were upregulated in each tissue, a transcriptome prolife like that of a fully differentiated cell was not present. Importantly, these analyses also confirmed that PLC-mcoET3 present within the reproductive tissues was not due to contribution of these transplanted cells to the germline during fetal development.

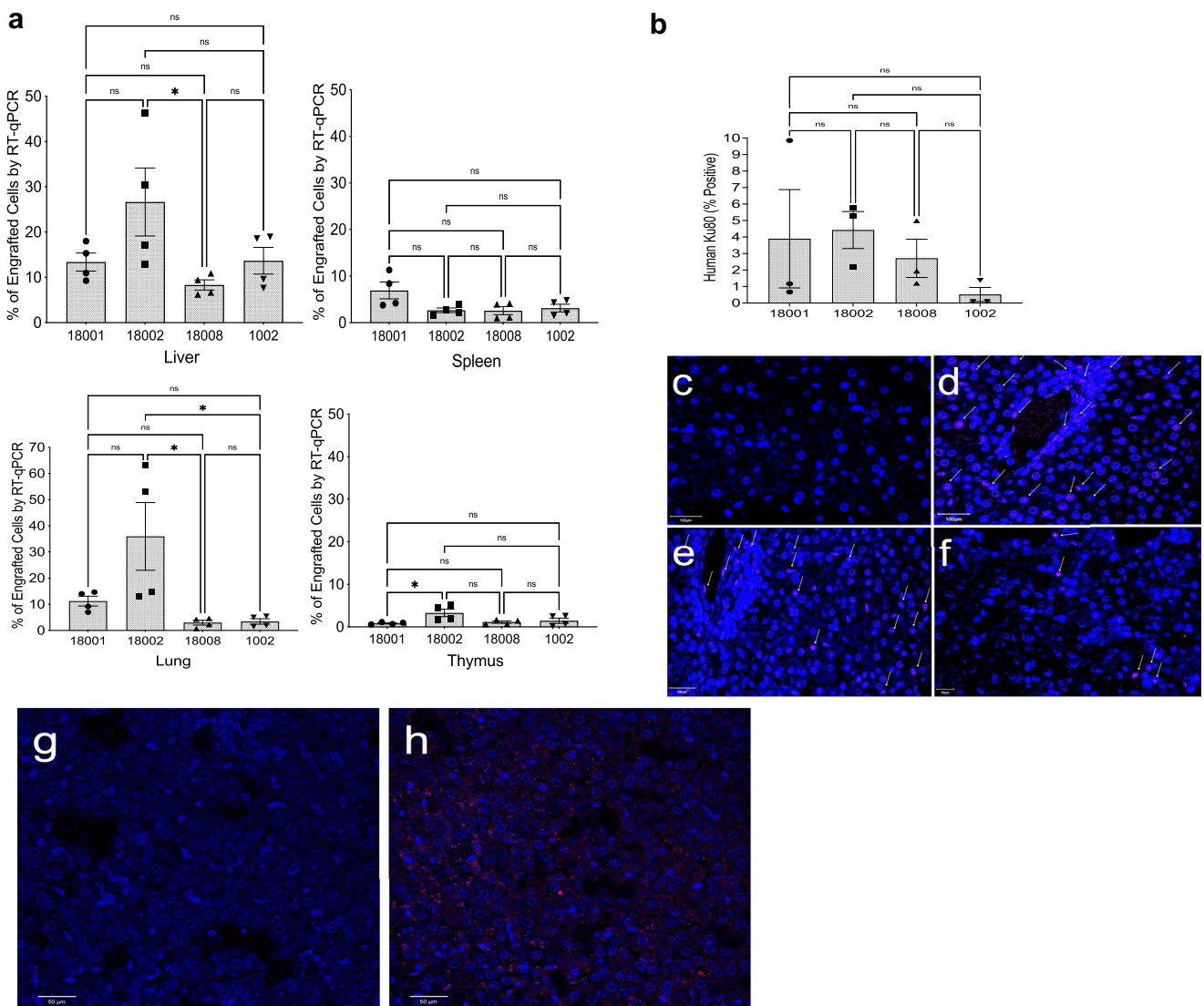

**Fig. 5 | Human PLC-mcoET3 engraft in multiple tissues after IUTx. a** RT-qPCR (in triplicate) (*n* = 4 animals, *n* = 4 experiments) using mcoET3-specific primers on RNA isolated from different organs. Animals had similar levels of engrafted PLC-mcoET3 in all organs analyzed; 18,002 had statistically significant higher levels than 18,008 in the liver, lung, and thymus; **b** liver quantification of human Ku80-positive cells by immunohistochemistry confirms the presence of donor PLC cells in all animals (with less sensitivity than by RT-qPCR); **c**–**f** representative images of Ku80 staining in liver (*n* = 3 animals) (white arrows = positive cells); **c** control non-transplanted animal; **d**–**f** IUTx animals (*n* = 3 animals); **g** isotype control and **h** presence of ET3/FVIII-producing cells between the hepatocytes and interlaced with the hepatic sinusoids in IUTx HA animal after anti-human FVIII staining. Since HA animals with this mutation lack factor VIII antigen in all tissues (cross-reactive material negative), the presence of ET3/FVIII detected results from the production of ET3/FVIII by the transplanted cells. Images were acquired with an Olympus Fluoview FV1000 confocal microscope with Olympus UPlanFLN- 40×/1.30 oil objective (scale bars: **c**–**e** = 100 μm; **f**–**h** = 50 μm). Percentage of positive cells for the different markers was calculated after counting a minimum 1500 DAPI+ nuclei (*n* = 3 experiments). Data are shown as Mean ± SEM. One-way ANOVA followed by Tukey's multiple comparison test was used to determine significant differences ($p \leq 0.05$ was considered significant). Liver *$p$ = 0.0406; Lung *$p$ = 0.0182 (18002-18008) and *$p$ = 0.0199 (18002-1002); **a** ns $p$ values in sequence from left to right: liver $p$ = 0.1685; 0.8285; >0.999; 0.1818; 0.8041; spleen $p$ = 0.0763; 0.0702; 0.1244; >0.9999; 0.9906; 0.9854; lung $p$ = 0.0845; 0.8127; 0.8366; >0.9999; thymus $p$ = 0.9818; 0.8695; 0.0755; 0.144; 0.9785. Source data are provided as a Source Data file.

Immunohistochemistry using antibodies specific for selected human proteins encoded by the upregulated transcripts was also performed in liver sections of the animals that displayed the highest number of those human liver transcripts, and representative images are shown in Figs. 7 and 9. A representative image of LYVE-1 positive cells within the endothelium of the liver sinusoids can be seen in detail in one of the transplanted animals (Fig. 9c, d). While none of the hepatocytes in the slides analyzed are positive for human albumin (Fig. 7h–k), some hepatocytes were found to be positive for the urea cycle enzyme CPS1, and representative images are shown in Fig. 9e–i.

## Discussion

The excitement for treating genetic disorders during gestation has been reawakened by the development of novel gene transfer and gene editing technologies, the better understanding of fetal development and fetal-maternal interactions, the advances in safety of prenatal manipulations, and the ability to produce and use pre/peri-natal-derived cells and/or gene-modified cells[16–23].

While IUTx clinical trials have been initiated to treat monogenic disorders in which the fetuses are at risk of demise and/or threatened by the development of irreversible pathologies, other diseases that would be ideal to treat in utero are those in which a minimal correction would

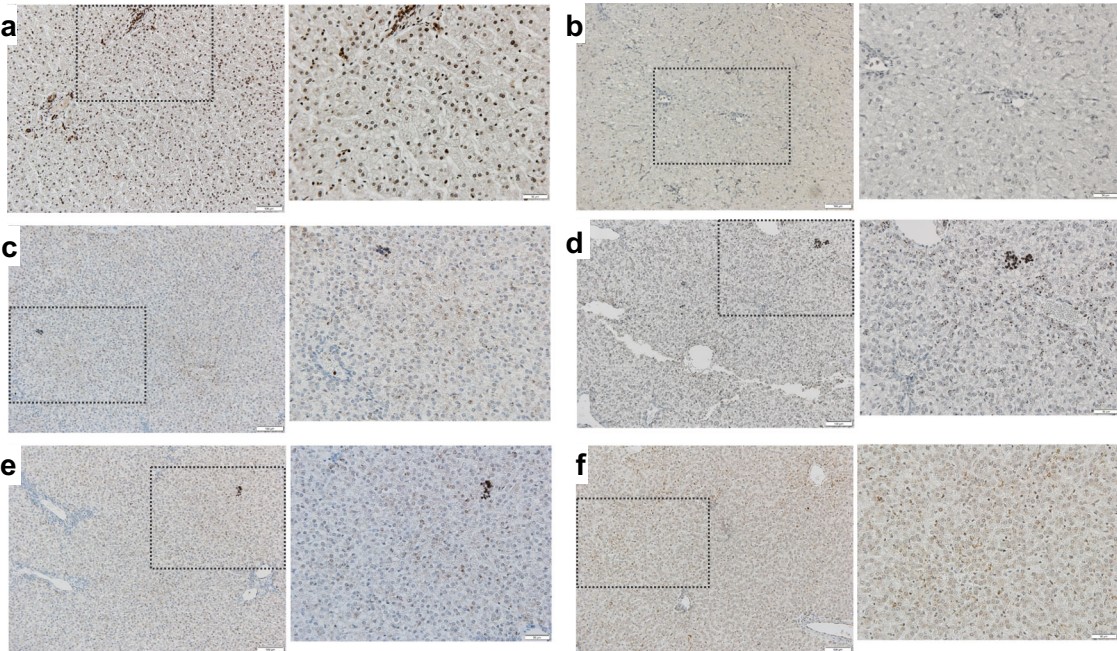

**Fig. 6 | Detection of Human Ku80 (nuclear antigen)-positive cells in liver of IUTx animals by immunohistochemistry confirms the presence of donor PLC-mcoET3 cells without evidence of histomorphological alterations or clonal proliferation at the sites of PLC-mcoET3 engraftment. a** Representative images (left: scale bar = 100 μm and right: scale bar = 50 μm) of staining for the human-specific nuclear protein Ku80 (dark brown) in human liver sections; **b** Representative images (left: scale bar = 100 μm and right: scale bar = 50 μm) of control non-transplanted sheep demonstrating the absence of human cells; **c–f** Representative images (left: scale bar = 100 μm and right: scale bar = 50 μm) of staining for the human-specific nuclear protein Ku80 in IUTx animals (*n* = 4 animals,

each panel a different animal), demonstrating the presence of human cells. The dark nuclei of the human cells can be seen within the parenchyma of the different liver sections, without displaying signs of disruption of the liver cellular architecture. Slides from all test animals and negative controls were analyzed in at least three different experiments. To eliminate observer bias, an ImageJ script was utilized to process three different regions of interest (ROIs) and identify positive cells. The script deconvoluted each ROI image into an H&E image and a DAB image, then considered positive cells using size, circularity, and color thresholding. All Images were acquired LEICA DM4000B brightfield microscope using 10× and 20× objectives.

substantially impact the future quality of life (QOL) of the affected individuals, and those in which it is possible to exploit the immunologic environment at the time of treatment to prevent postnatal immune reaction to transgene-encoded proteins[6,9,17,24]. It is also critical that prenatal genetic diagnosis is available and reliable, and that the correlation between genotype and phenotype can predict long-term clinical prognosis[6,9,17,24]. Thus, HA is an ideal disease to correct prenatally, since 70% of PHA have a family history of the disease, reliable prenatal diagnosis is available, and the type of mutation in combination with family history can predict whether patients are prone to develop inhibitors. In addition, an increase of only 5% of FVIII in circulation can change a life-threatening spontaneous bleeding phenotype into a phenotype appropriate for mild physical activity, and an increase of 5–15%, can withstand higher-risk physical activity, in which bleeding events only occur after major trauma or surgery[8,9,17,25].

Although the newest FVIII products and therapeutic regimens available to PHA have greatly improved the standard of care and the QOL for many, there are still unmet medical needs requiring the development of novel therapeutic approaches to address the challenges and the gaps in the standard treatments[2,26,27]. Importantly, the goal of a cure remains unrealized. For instance, even with the best prophylactic treatments, breakthrough bleeding still occurs, contributing to long-term morbidity. Treatment burden, especially in pediatric patients, is a reality, as even with subcutaneous routes of administration, treatments are still invasive, both physically and psychologically[2]. In addition, prophylactic treatment is lifelong, and >30% of PHA develop FVIII inhibitors[28,29]. Importantly, both persons with severe[30] and non-severe HA[31] who develop inhibitors have an increased mortality risk. Also of note is that some patients can experience spontaneous bleeding starting during the neonatal period

or infancy. Reports show that 9.5% of HA newborns need replacement treatment within the first 24 h after birth, and 44% have a bleeding episode by 1 mo.[26,27] Moreover, infants suffer cranial (24%), oral (30%), soft tissue (7%), and joint (16.2%) bleeds[26,27]. Thus, even a 5% increase in FVIII activity during the neonatal period could avert early bleeding events. Therefore, these patients could greatly benefit from early intervention.

Clinical trials in adults using AAV gene therapy (AAV-GT) demonstrate the promise of a cure in adulthood, or possibly late adolescence, with one treatment, but the decreasing FVIII levels over time suggest additional treatments will be needed[32]. Pre-existing AAV immunity can be as high as ~60%, depending on donor age and AAV serotype, severely impacting PHA's ability to receive such therapies. Inflammatory responses post-AAV-GT cause loss of therapeutic effect, and AAV's episomal nature makes pediatric application of AAV-GT suboptimal, since hepatocyte proliferation during liver growth could lead to dilution and loss of these liver-targeted therapies, while re-administration is compromised by the development of immunity to AAV components[33]. In addition, when AAV vectors to deliver FIX and X were administered in early gestation wild-type macaques, the sustained expression of the coagulation factors was correlated with AAV vector integration, albeit in the absence of clinical toxicity[18]. Thus, delivering cells during gestation that proliferate physiologically in vivo and secrete clinically therapeutic levels of FVIII is a promising and safe approach for providing long-term/permanent correction of HA[6], as transducing cells in vitro allows safeguards in production that are not possible with direct vector injection and eliminates the risk of inadvertently transducing undesired tissues/cells, e.g., those of the germline.

IUTx has been performed safely in human patients, with many patients receiving three injections spaced 1 week apart, proving that

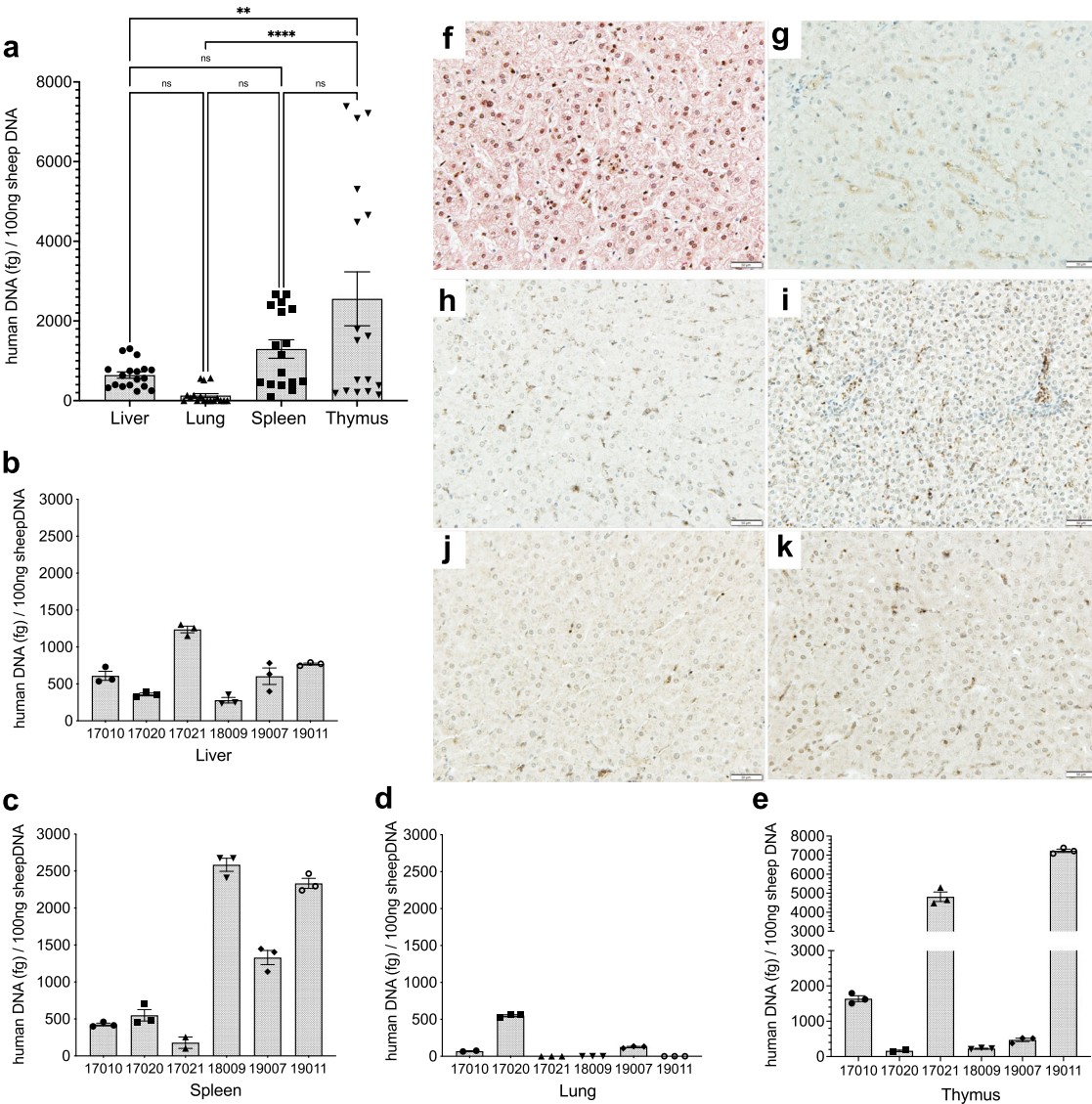

**Fig. 7 | PLC-mcoET3 persist long-term in the engrafted tissues. a–e** qPCR using human Alu-specific primers performed in triplicate on DNA extracted from liver, lung, spleen, and thymus ($n = 6$ animals) at 2.3–5 years post IUTx; amount of human DNA extrapolated from a standard curve consisting of sheep DNA with (0.25, 0.5, 1, 5, and 10 ng of human DNA from PLC to a total of 100 ng of DNA per sample) or without human DNA. All the tissues analyzed contained human DNA; the same tissue samples from control non-treated animals showed no target amplification; **b–e** Individual data of human DNA content found in organs of individual animals depicted in (**a**); **f–k** Representative images of liver tissue sections confirming the presence/location of human cells by IHC with antibodies against the human nuclear protein, Ku80, (dark brown) and human albumin (red); **f** Human liver tissue section demonstrating the positive dark nuclei and cytoplasmic albumin in red; **g** Non-transplanted sheep liver control showing absence of dark human nuclei or cytoplasmic human albumin. **h–k** Liver tissue sections from IUTx animals ($n = 4$ animals), dark human nuclei can be seen interspersed between hepatocytes nearby liver sinusoids, while none of the hepatocytes in the slides analyzed are positive for human albumin. Slides from all positive and negative controls were stained in three different experiments and test animals were analyzed in at least three distinct ROIs using ImageJ scripts. All Images were acquired on a LEICA DM4000B brightfield microscope using a 20× objective. Data are shown as Mean ± SEM. One-way ANOVA followed by Tukey's multiple comparison test was used to determine significant differences and $p \leq 0.05$ was considered significant. \*\*$p = 0.0016$, \*\*\*\*$p = <0.0001$, $p = 0.7316$ liver-lung; $p = 0.5619$ liver-spleen; $p = 0.1072$ lung-spleen; $p = 0.0713$ spleen-thymus. Source data are provided as a Source Data file.

accessing the early human fetus multiple times, using a minimally invasive, ultrasound-guided approach, poses minimal risk[34]. Importantly, during fetal life, activation of FX occurs predominantly via tissue factor activity, making it largely independent of the FIXa/FVIIIa phospholipid complex[35]. As such, the fetus develops without hemorrhage, even if little or no FVIII or FIX are present[35–37]. The unique hemostasis of the fetus should thus allow IUTx to be performed safely for HA, as demonstrated in a single case study reporting the successful outcome of IUTx in a human fetus with severe HA, without causing bleeding[38]. Data also suggest that the recipient was tolerized to FVIII[38], but no follow-up on this patient was ever published.

Remarkably, is that in contrast to hemophilia B, very few animal studies have been designed to test the efficacy and safety of prenatal therapies for HA. In one study, an adenoviral vector encoding FVIII was injected into mice in utero. Therapeutic levels were seen at day 2 of life, but FVIII levels had dropped by sevenfold by day 15, and were gone by day 21[39]. Recently, other investigators performed IUTx at embryonic day 14.5 in wild-type murine fetuses using FVIII-transduced placental cells, but evaluation of the recipients was only performed by GFP expression, and levels of FVIII in circulation or immunological consequences of the approach were not investigated[40].

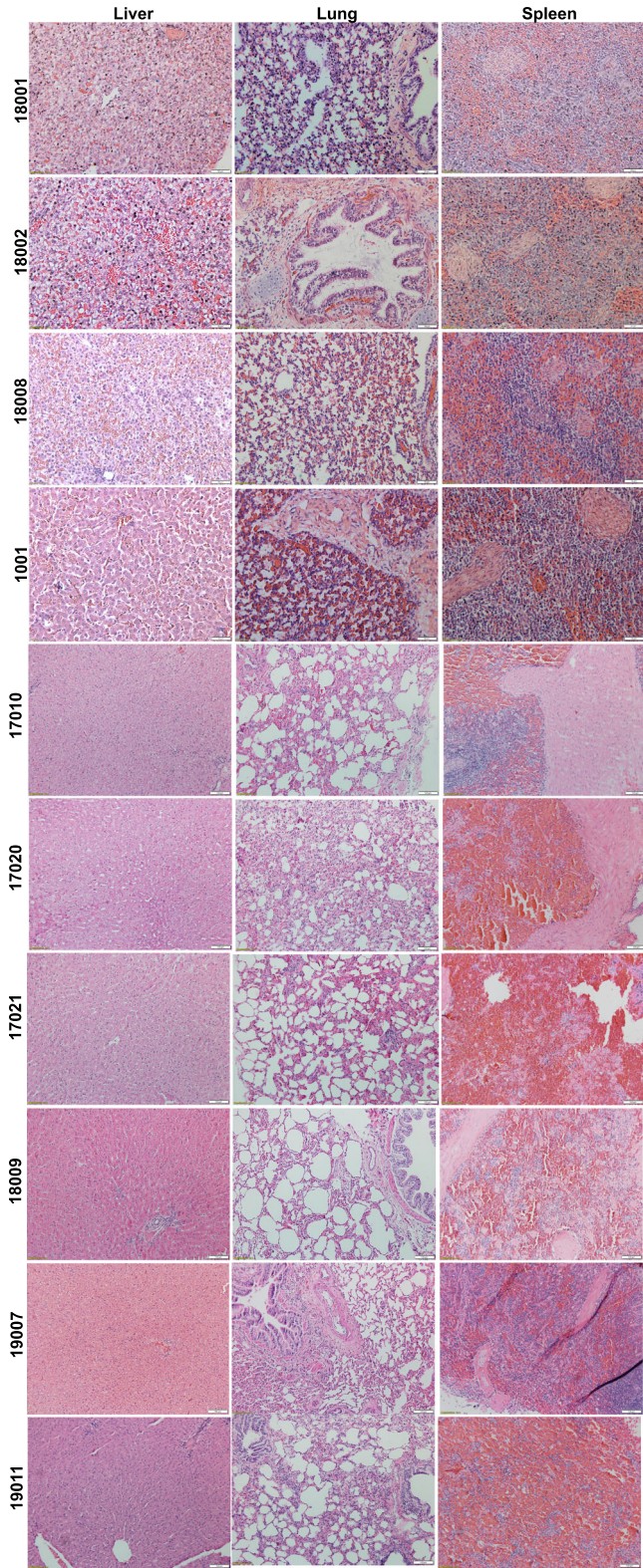

**Fig. 8 | No evidence of any lentiviral-related or procedural toxicity after histopathological evaluation of tissues from IUTx animals.** Representative images of H&E staining of tissue sections from IUTx animals ($n = 10$ animals) demonstrating no changes in tissue architecture, or tumorigenesis. All images were acquired with a Leica DM4000 B microscope using a 10× objective.

Here, we report long-term data on safety and efficacy of delivering cells engineered to secrete fVIII/ET3, at clinically relevant doses, in a large animal model of IUTx. Because the weight of the fetal sheep at the time of IUTx is comparable to that of humans early in gestation, and in adulthood it equals or surpasses that of humans, no scale-up in cell dose would be necessary if the translation to the clinic of these therapies is deemed to be safe. In addition, the similarity between total plasma volume between human and sheep ensures that the long-term therapeutic levels of FVIII activity found in plasma of IUTx animals after birth would be comparable to those in humans, as both humans and sheep undergo exponential increases in total plasma volume from fetal life to adulthood. The data in this study show that administration of PLC-mcoET3 at the equivalent of 16–18 g.w. in humans resulted in elevated plasma FVIII activity that exceeded that of control non-IUTx animals by $>48.4 \pm 12.3\%$, for >3 years after birth. The FVIII activity levels determined by FVIII activity assays were also confirmed by LC-MS, which, in addition, demonstrated that ET3 protein was present in the plasma of treated animals. Furthermore, in seven out of eight animals, FVIII levels at 1-year post-IUTx did not differ significantly from those at 3 years post-IUTx, and the one animal in which levels of FVIII declined still maintained FVIII levels >5% above normal. Interestingly, higher plasma FVIII levels were not detected in animals that received higher cell doses, or cells with higher FVIII secretion. Importantly, none of the treated animals developed anti-FVIII/ET3 IgM or IgGs, or ET3-specific $T_{h1}$ or $T_{h2}$ cells. This is in contrast to what was reported in another study using juvenile sheep that received the exact same therapy (PLC-mcoET3) by the same administration route (IP), and in which 66% of animals developed anti-FVIII/ET3 IgGs and low titers of anti-FVIII and anti-ET3 inhibitory antibodies[13].

Also of interest is that IUTx-treated recipients did not develop immunity or anti-HLA antibodies to the transplanted human PLC, but they maintained robust reactivity to foreign antigens. Furthermore, hematological parameters and liver enzymes were normal, proving the absence of any liver toxicity. PLC-mcoET3 were found to lodge in all major organs analyzed, without leading to histopathological alterations, and evaluation of mcoET3 expression by RT-qPCR and human genomic DNA by qPCR in the same tissues not only confirmed the long-term engraftment of the transplanted cells, but also demonstrated the continued transcription of the transgene.

Of note is that RNA-seq analysis to identify changes in engrafted PLC-mcoET3's transcriptome from prior to transplant to that of a transcriptome comparable of that of an organ-specific cell type demonstrated that although several transcripts were upregulated, these cells never expressed sets of genes matching those of differentiated cells. Since the liver in sheep weighs approximately 1Kg, it would be impossible to absolutely exclude the possibility of the existence of fully differentiated cells. Nevertheless, is important to note that these same regions contained cells positive for the human nuclear protein Ku80. Collectively, the data shows that transplanted PLC-mcoET3 express some transcripts and produce some of the proteins of the tissue where they lodge, but cannot confirm the presence of a fully functional differentiated organ-specific cell.

The study also shows that within the samples analyzed, there were significant differences in levels of PLC-mcoET3 engraftment in the different organs and between animals. Even if multiple samples from each animal/organ were analyzed, due to the large volume of the organs, it difficult to ascertain whether it is possible to extrapolate these results to the totality of the organs. Thus, the best readout for the outcome of the therapy are the levels of FVIII activity detected in the plasma of these animals, which can truly measure the overall output of FVIII by the engrafted cells.

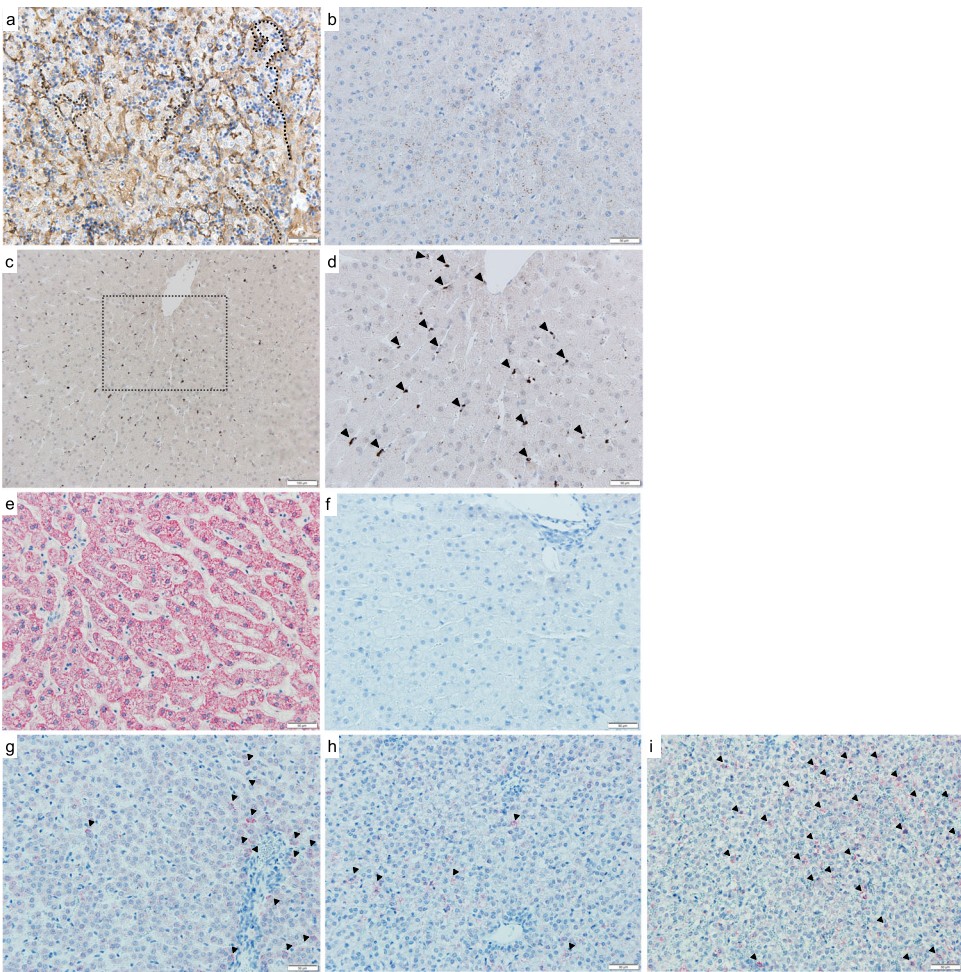

**Fig. 9 | Targeted evaluation of liver-specific cell markers by immunohistochemistry demonstrates the expression of human markers in some of the engrafted cells. a** Representative image of LYVE-1 staining of human liver tissue section demonstrating liver sinusoidal endothelial cells (LSECs) stained in dark; for reference some of the sinusoids are annotated with dotted lines (scale bar = 50 μm); **b** Representative image of LYVE-1 staining of non-transplanted sheep control liver tissue section, confirming the absence of human LSECs (scale bar = 50 μm); **c** Representative image of LYVE-1 staining of an IUTx sheep liver tissue section in which transcripts for LYVE-1 were found (n = 5 animals analyzed) demonstrating the presence of human LSEC-like cells expressing LYVE-1 (scale bar = 100 μm); **d** magnification of the dotted rectangle (scale bar = 50 μm) in (**c**) showing in detail the LYVE-1+ cells which are marked with black arrowheads; (**e–i**) Representative images of human urea cycle enzyme CPS1 (pink) staining of liver tissue sections (scale bars = 50 μm); **e** Human liver tissue section with hepatocytes positive for CPS1 (in pink); **f** Non-transplanted sheep control liver tissue section, confirming the absence of hepatocytes expressing human CPS1; **g–i** Representative images of liver tissue sections from IUTx animals in which transcripts for CPS1 were found (n = 4 animals) showing several hepatocytes positive for human CPS1. Slides from all positive and negative controls and test animals were analyzed at least in three distinct areas in three separate experiments. All Images were acquired with a LEICA DM4000B brightfield microscope.

Attesting to the safety of the procedure and therapy is also a main goal, and in this study animals that were lost to the study were due to causes unrelated to treatment or IUTx procedure. This includes the late-term abortions in HA cloned animals, since this problem has been well-documented after this type of assisted reproductive technology, and was not due to the transplant itself[41]. Unfortunately, because when these studies were initiated the carriers in the HA sheep colony were aged, and cloning resulted in miscarriages, we were not able to produce many HA animals by natural breeding. The one HA animal treated by IUTx was stillborn because of labor and delivery complications. However, despite the traumatic birthing process, at necropsy, the animal did not display evidence of the abnormal bleeding that is seen in this line of HA animals[14,15], and ET3/FVIII-producing cells were identified in the liver and thymus of this animal.

Thus, these studies attest to the feasibility, immunologic advantage, and safety of treating HA prior to birth. Of significance is that IUTx with PLC-mcoET3 could eliminate the need for weekly dosing, while maintaining therapeutic levels which exceeded the new suggested clinical objective of >15% FVIII, which is intended to provide effective protection from the majority of spontaneous bleeds[25]. Since at 3 years after IUTx, the average plasma levels of FVIII were 48.4 ± 12.3% above the control, this approach could also provide enough factor to control other acute bleeds, in particular, subclinical bleeds and those associated with traumas, surgery, and/or intense physical activity[25]. Nevertheless, confirmation of phenotypic correction by evaluating whether the annualized bleeding rate is significantly reduced from baseline throughout the lifetime of HA animals, and evidence that an immune response is still not present in these animals is a must to ascertain the true clinical impact of this prenatal approach.

## Methods
### Ethical statement
All research was performed in accordance with relevant guidelines and regulations and animal studies approved by the Wake Forest University School of Medicine Institutional Animal Care and Use Committee (IACUC). Discarded human placentas were obtained from

full-term deliveries after written informed consent and compensation for their time according to guidelines from the Office of Human Research Protection and approved by the Wake Forest University School of Medicine Institutional Review Board (IRB).

## Study design and prenatal transplantation

All procedures involving animals were approved by the Wake Forest University School of Medicine Institutional Animal Care and Use Committee. All experimental sheep are an outbred mix of Alpine White, Suffolk, Rambouillet, and Merino strains and are housed in fenced pastures with access to shelters and/or stalls at the Wake Forest University School of Medicine large animal facility under the veterinary care of the Animal Resources Program (ARP). Sheep are provided with water, appropriate food for the season and pregnancy status, and supplements such as vitamin E, selenium, and calcium. Lambs receive the mother's colostrum or artificial colostrum and are bottle-fed when needed. Sheep are vaccinated and examined regularly per ARP veterinarians and/or veterinary technicians to ensure the health and nutrition status of the flock. Fetal sheep ($n = 25$), males and females, were injected with PLC-mcoET3 intraperitoneally at 59–65 g.d. (full-term gestation: 145–150 d), which corresponds to ~16–18 weeks of gestation in humans, by ultrasound-guided transabdominal percutaneous injection using a 22-gauge, 152 mm echogenic spinal needle (Hakko Co, 6PTC22) as previously reported[42], at the cell doses of $10^7$–$4 \times 10^8$ cell/kg in 500uL of QBSF-60 (Quality Biologicals Inc. NC0823508). Although HA is an X-linked disorder and predominantly affects males, here the product was administered to both males and females. Of the 25 transplanted animals, 13 were available for long-term evaluation (up to 1 year) and eight of these animals were followed up to 3 years; four were only evaluated shortly after birth (~3 months post-transplant), while eight were lost to study. All deceased/euthanized animals had medical complications unrelated to procedure and or product administration. When evaluating treatment efficacy, tests were run blind to animal treatment status. To ensure statistical significance, animal numbers were determined by power analyses using G*Power v3.1.9.2.

Additional information pertaining to study design, all research methods, and outcomes can be found in Supplementary Table 1 and Supplementary Materials.

**Isolation, culture, and transduction of placental cells.** Placental cells were isolated, cultured, transduced with a lentiviral vector encoding mcoET3, a human myeloid-codon-optimized, bioengineered FVIII transgene containing high expression elements from porcine FVIII[12], and analyzed for FVIII production, phenotype, viability/function, genomic stability, and immunogenicity as previously described[11] and detailed in Supplementary Table 1. PLC-mcoET3 used in this study met the criteria of having a VCN < 5 and production of FVIII /10^6 cells/24 h > 5 IU.

**Factor VIII (FVIII) activity assay.** Blood was collected in sodium citrate tubes (BD Biosciences, 369714), and plasma was isolated by centrifugation at $400 \times g$, 30 min, and stored at −80 °C until aPTT assays were executed. aPTT assays were performed by the Wake Forest Baptist Medical Center Special Hematology Laboratory using standard clinical procedures. In brief, standard reference plasma was produced by pooling plasma from a large number (e.g., 40) of normal healthy volunteers, and was assigned a value of 100% FVIII levels. Serial dilutions of this reference plasma were then prepared in FVIII-deficient plasma to establish a standard curve with values of 100, 50, 33, 25, 12.5, 10, and 1% of normal FVIII levels. This normal human reference plasma, plasma from transplanted animals, and plasma pooled from 4–6 non-transplanted control sheep to be assayed were mixed with the FVIII-deficient plasma, and the standards and samples were run on an ACL Top 300 CTS clinical coagulometer (Instrumentation Laboratories,

00000280060). The results were then plotted on Log-Lin paper, plotting dilutions on the logarithmic $X$-axis and clotting times on the linear $Y$ axis. A vertical line was drawn from the dilution that yielded 100 IU/dl (100%). Where this line intercepted the line of best fit for the reference plasma standard—a line at a right angle to this was drawn until it intercepted the supernatant sample being assayed. A vertical line was then dropped until it intercepted the $X$-axis to establish the FVIII level in the normal and transplanted sheep.

All tests were run blind to animal treatment status. Percentage increase in FVIII activity over a pool of normal control sheep was calculated using Eq. (1).

$$\%FVIII\ Activity\ Increase$$
$$= \left( \frac{(\%\ FVIII\ Activity\ in\ animal\ receiving\ therapy\ -\ \%FVIII\ Activity\ normal\ controls)}{(\%FVIII\ Activity\ normal\ controls)} \right) \times 100$$
$$(1)$$

## Proteomic analysis

To confirm the presence of ET3 and hFVIII in plasma of transplanted animals, protein digestion liquid chromatography-tandem mass spectrometry (LC-MS) was performed in a blinded fashion as detailed below. The scaffold DIA normalization algorithm for intensity quantification was applied to appropriately adjust values for comparison purposes. Summed intensities from different MS samples were first log10 transformed and a histogram is built for each sample. Quarterlies for each sample were calculated and compared to the quarterlies of the entire experiment. The exclusive intensity represents the summarized intensity value of only the peptides that are associated with this protein.

## Protein digestion

Protein content from plasma samples was estimated using the fluorescence-based protein assay EZQ (Invitrogen #R33200). 100 μg protein extracts were reduced, alkylated with iodoacetamide, and digested with a trypsin/Lys-C protease mixture using Thermo Scientific EasyPep Mini MS Sample prep kit (Cat #A40006), following the provided protocol. The trypsin/Lys-C was added 1:10 (enzyme: protein) for digestion. Samples were cleaned up for analysis using the column provided with the kit and reconstituted in 100 μl 0.1% formic acid in water, making a final concentration of 1 μg/ul for analysis.

## Liquid Chromatography

Samples were analyzed using an UltiMate 3000 RSLCnano system (Thermo Scientific, San Jose, CA). The peptides were trapped prior to separation on a 300 μm i.d. × 5 mm C18 PepMap 100 trap (Thermo Scientific, San Jose, CA) for 5 min at 10 μl/min. Separation was performed on a 50 cm uPAC C18 nano-LC column (PharmaFluidics, Ghent, Belgium) on an EasySpray source (Thermo Scientific, San Jose, CA) fitted with a 30 μm ID stainless steel emitter (PepSep, Marslev, Denmark). Separation was performed at 350 nl/min using a gradient from 1–45% for 60 min (Solvent A: 0.1% Formic Acid; Solvent B: Acetonitrile, 0.1% Formic Acid).

## Mass spectrometry

Data Independent Analysis (DIA) was performed using an Eclipse Tribrid Orbitrap mass spectrometer (Thermo Scientific, San Jose, CA)[43,44] with a chromatographic library as described by Searle et al.[45]. Briefly, six gas phase fractions (GPF) of the biological sample pool were used to generate a reference library. The GPF acquisition used 4 $m/z$ precursor isolation windows in a staggered pattern (GPF1 398.4–502.5 $m/z$, GPF2 498.5–602.5 $m/z$, GPF3 598.5–702.6 $m/z$, GPF4 698.6–802.6 $m/z$, GPF5 798.6–902.7 $m/z$, GPF6 898.7–1002.7 $m/z$) at a resolution of 60,000, AGC target was set to custom with a normalized target of 1000%, maximum injection time was set to dynamic with a minimum of nine points across the peak, and an NCE of 33 using higher-energy collision dissociation (HCD). Biological samples were run on an identical gradient

as the GPFs using a staggered window scheme of 8 $m/z$ over a mass range of 385–1015 $m/z$. Precursor isolation was performed in the Orbitrap at 60,000 resolution with a dynamic maximum injection allowing for a minimum of nine points across the peak, a custom AGC normalized to 1000% and an NCE of 33 using HCD. The species-specific FASTA database for Ovis aries (UP000002356) containing 23,111 proteins was downloaded from Uniprot. An empirically corrected library that combines the GPF and the deep neural network Prosit[46] was used to generate predicted fragments and retention times. The ET3 sequence was added to the Ovis aries UP000002356 FASTA, converted to a Prosit compatible CSV format and submitted to Prosit. The converted Prosit file was then converted to the EncyclopeDIA DLIB format. ScaffoldDIA (Proteome Software, Portland, OR) using the Ovis aries UP000002356 DLIB produced an empirically corrected library using the GPF-DIA injections. DIA-MS samples were analyzed using Scaffold DIA (3.1.0). DIA-MS data files were converted to mzML format using ProteoWizard (3.0.19254)[47]. Deconvolution of staggered windows was performed. Samples were aligned based on retention times and individually searched against Sheep_ET3.elib with a peptide mass tolerance of 10.0 ppm and a fragment mass tolerance of 10.0 ppm. Variable modifications considered were: Carbamidomethylation C. The digestion enzyme was Trypsin with a maximum of 1 missed cleavage site. Peptides with charges from 2–3 and length of 6–30 amino acids were considered. Peptides identified in each sample were filtered by Percolator (3.01.nightly-13-655e4c7-dirty)[48–50] to achieve a maximum FDR of 0.01. Individual search results were combined and peptide identifications were assigned posterior error probabilities and filtered to an FDR threshold of 0.01 by Percolator. Peptide quantification was performed by Encyclopedia (1.2.2). For each peptide, the 5 highest quality fragment ions were selected for quantitation. Proteins that contained similar peptides and could not be differentiated based on MS/MS analysis were grouped to satisfy the principles of parsimony. Protein groups were thresholded to achieve a protein FDR <1.0%.

### Liver enzymes tests, white blood cell (WBC) counts, and hematocrit levels

Aspartate aminotransferase (AST) and alanine aminotransferase (ALT) blinded assays were performed by Cornell University Animal Health Diagnostic Center. Normal values for AST and ALT in sheep are <280 and <50 U/L respectively. Whole blood was used to determine WBC/mL using the LeukoCheck test kit (Biomedical Polymers 21243285). Hematocrit levels were determined by the veterinary staff.

### Anti-mcoET3 IgM and IgG antibody enzyme-linked immunosorbent assay (s) (ELISA)(s)

High-binding plates (Corning, 9018) were coated with ET3 protein as previously described[51]. Sheep plasmas were serially diluted starting at 1:20 dilution, and antibody binding was detected with anti-ovine IgG:ALP (Bio-Rad STAR88A) or anti-ovine IgM:ALP (AbCam, AB112761), and p-nitrophenyl-phosphate (Bio-Rad, 1721063). The antibody titer was defined as the dilution of plasma with an absorbance value > 2 SD above the mean OD from control sheep plasma ($n$ = 4) Supplementary Fig. 7a, b.

### Quantification of mcoET3-PLC engraftment by RT-qPCR

To quantify levels of engraftment of mcoET3-PLC in IUTx recipients, RT-qPCR was performed using Applied Biosystems QuantStudio 6 Flex Real-Time PCR system. Tissue samples from IUTx recipients were collected from liver, lungs, thymus, spleen, brain, kidneys, pancreas, and intestine, and preserved in RNAlater (Qiagen, AM7020). RNA was isolated and converted to cDNA using the Omniscript RT kit (Qiagen, 205113) as per manufacturer's instructions. RT-qPCR was performed using 10 ng cDNA/sample, the TB Green Advantage qPCR Premix (Takara Bio, 639676) and primers specific for sheep GAPDH (housekeeping gene) and mcoET3 (Fwd Primer: GCAGGGATGTCCACCA

CATT; Rev Primer:GTATGGACAGTGGGCACCAA) at final concentrations of 200 nM and 350 nM, respectively. The raw Ct values obtained from the tissue samples were normalized using the Ct value for the housekeeping gene to obtain ΔCt. Percentage of engraftment was determined by comparing ΔCt to a standard curve consisting of increasing percentages of mcoET3-PLC (0, 0.01, 0.1, 1, 5, and 10%) in sheep stromal cells (MSCs).

### Quantification of mcoET3-PLC engraftment by Alu qPCR

To quantify levels of mcoET3-PLC present in different organs, qPCR was performed as previously described[52] using the Applied Biosystems QuantStudio 6 PCR system. Tissue samples were collected from liver, lung, spleen, and thymus, and preserved in Allprotect Tissue Reagent (Qiagen, Catalog No. 76405). DNA was isolated using the DNEasy Blood and Tissues kit (Qiagen, Catalog No. 69504), as per manufacturer specification. qPCR was performed using 100 ng gDNA per sample, with TaqMan Universal Master Mix (ThermoFisher, Catalog No. 4440043), and primers designed for the Human Alu target (Forward Primer: 5′-GGTGAAACCCCGTCTCTACT-3′) (Reverse Primer: 5′-GGTTCAAGCGATTCTCCTGC-3′), and a hydrolysis probe (5′-CGCCCGGCTAATTTTTGTAT-3′) labeled with reporter 6-FAM and quencher BHQ1. Primer working concentration and hydrolysis probe working concentration were 0.2 and 0.25 μM, respectively. A standard curve was prepared consisting of sheep DNA without human DNA, or sheep DNA spiked with increasing amounts, (0.25, 0.5, 1, 5, and 10 ng) of human DNA derived from PLC to a total of 100 ng of DNA per sample. Each sample was evaluated in triplicate under the following PCR conditions; 1 cycle of 95 °C for 10 min, followed by 50 cycles of 95 °C for 15 s, 56 °C for 30 s, and 72 °C for 30 s. A No-Template Control was included for each run.

### RNA-seq library construction and Illumina NGS.
RNA extracted from (1) an aliquot of the mcoET3-PLC that were transplanted, (2) from the different tissues of sheep that were transplanted in utero, and from (3) same tissues from control non-transplanted sheep was quantified using a NanoDrop 2000 (Thermo Fisher Scientific, Inc.), and RNA integrity was assessed using the Bioanalyzer RNA 6000 Nano assay and 2100 Bioanalyzer (Agilent). All RNA samples were then sent to Qiagen for RNA-seq library construction and Illumina NGS. Complete transcriptome RNA-seq libraries were created using the QIAseq UPXome RNA Lib Kit HMR (Qiagen Catalog No. 334705) following the manufacturer's standard protocol. In brief, total RNA was subjected to ribosomal RNA (rRNA) suppression using QIAseq FastSelect rRNA removal kits HMR (Qiagen Catalog No. 334386) and then reverse transcribed into cDNA using random hexamers and oligo-dT primers for 90 min at 42 °C. During cDNA synthesis each sample receives a unique sample ID that allows for cDNA pooling and all subsequent library construction steps of pooled samples was done in a single tube to minimize batch effects. Following cDNA pooling, two rounds of bead selection/reaction clean-up was done with QIAseq beads (Qiagen Catalog No. 1107149). The cDNA was then amplified with unique-dual sample indexes using the QIAseq UX 12 Index Kit IL UDI (Qiagen Catalog No. 331801).

Completed libraries were then purified using two rounds of bead-based cleanup with QIAseq beads. Final libraries were quality controlled using an Agilent Bioanalyzer and quantitated by real-time qPCR using the QIAseq Library Quant Assay Kit (Qiagen Catalog No. 333314) and sequenced using an Illumina Next-Seq 500 using 75 base paired end sequencing using NextSeq 500/550 Mid Output Kit (150 Cycles).

### Data analysis: demultiplexing, read mapping.
FASTQ files were uploaded to QIAGEN RNA-seq Analysis Portal (https://geneglobe.qiagen.com/us/analyze/) and aligned to Homo sapiens (GRCh38.noalt.103) with analysis workflow version: 1.0. (https://resources.qiagenbioinformatics.

com/manuals/rnaanalysisportal/current/index.php?manual=QIAseq_UPXome_RNA_Lib_Kit.html).

RNA-seq libraries were aligned to Homo sapiens libraries after eliminating any possible cross-reactivity with ovine transcripts. We then compared the transcripts of the transduced mcoET3-PLC prior to transplant, with human transcripts known to be expressed in specific cells in different organs.

## Immunohistochemistry (IHC)

To determine the location of mcoET3-PLC in tissues, IHC staining was performed in paraffin-embedded tissues collected from the four animals analyzed at birth. Liver, lung, spleen, and thymus were stained using an antibody to the human-specific nuclear antigen, Ku80 (Cell Signaling Technologies, 2753S) and a goat anti-rabbit IgG Alexa-Fluor 594 (Life Technologies, A11012). To detect human FVIII/mcoET3 in a HA animal, a primary antibody targeting the C2 domain of human FVIII was used (Sekisui Diagnostics, ESH-8), and detected with a secondary antibody goat anti-mouse IgG AF-594 (Life Technologies, A11032). Positive tissue controls (human tissues) negative tissue controls (non-IUTx sheep tissues) were stained and evaluated in parallel with every experiment performed evaluating experimental tissue slides. Confocal images were acquired with an Olympus Fluoview FV1000 confocal microscope and an Olympus UPlanFLN- 40x/1.30 oil objective.

To identify PLC-mcoET3 in tissues and simultaneously visualize tissue morphology, IHC using chromogenic detection methods and antibodies to human-specific nuclear antigen, Ku80 (Cell Signaling Technologies, 2180), human urea cycle enzyme CPS1 (HepPar1) (Thermo Fisher, Catalog No. MSM4-966-P1), anti-human LYVE1 (AbCam, Catalog No. ab219556), and anti-human albumin (Thermo-Fisher, Catalog No. MA1-19174). Specifically, following tissue deparaffinization and rehydration, antigen retrieval (EDTA Buffer, pH 6.0) was performed using an IHC automated processor. Tissues were blocked with BLOXALL blocking solution (Vector Laboratories, SP-6000) and MAXBlock Blocking Medium (Active Motif, 15252) as per manufacturer instructions. Staining was performed using a rabbit anti-human Ku80 antibody (Cell Signaling Technologies, 2180), diluted in MAXBind-staining solution (Active Motif, 15253) and incubated overnight at 4 °C. The secondary antibody was a chicken anti-rabbit IgG antibody (Novus Biologicals, NBP1-75267), diluted in MAXBind-Staining solution and incubated 1 h at RT. VECTASTAIN Elite ABC reagent (Vector Laboratories, PK-6100) was used as per manufacturer instructions, followed by incubation in ImmPACT DAB reagent (Vector Laboratories, SK-4105) until color developed, and reaction was stopped with DI water. Slides were washed with water, and a hematoxylin counterstain was applied. To identify LYVE-1 or urea cycle enzyme CPS1 positive cells, tissue slides underwent antigen retrieval (Tris-based Buffer, pH 9.0) using an IHC automated processor, and blocking using BLOXALL Blocking solution (Vector Laboratories, SP-6000) and MAXBlock Blocking Medium (Active Motif, 15252) as per manufacturer specifications. Primary antibodies used were a rabbit anti-human LYVE1 (AbCam, ab219556), diluted 1:1000 in MAXBind-Staining solution (Active Motif, 15253) or mouse anti-human CPS1 (HepPar1) antibody (Thermo Fisher, MSM4-966-P1), diluted in MAXBind-Staining solution (Active Motif, 15253). The former was incubated overnight at 4 C, and the latter 1 h at RT as per manufacturer specifications. Secondary antibodies were, goat anti-rabbit IgG antibody conjugated to an HRP enzyme (AbCam, ab214880) at a pre-diluted, ready-to-use concentration, incubated 10 min at RT and anti-mouse IgG antibody conjugated to an AP enzyme (Vector Laboratories, MP-7714). After washing, ImmPACT DAB Peroxidase Substrate (Vector Laboratories, SK-4105) or ImmPACT Vector Red Substrate were added (Vector Laboratories, MP-7714), and incubated for 2 min at RT. Slides were immediately washed with water, and a hematoxylin counterstain was then applied. Tissue slides were coverslipped and imaged at 20X magnification using a LEICA DM4000B brightfield microscope.

To simultaneously detect human albumin and human nuclei, tissue slides underwent antigen retrieval (Citrate-based Buffer, pH 6.0), followed by IHC staining using the ImPRESS® Duet Double Staining Polymer kit (Vector Laboratories, MP-7714), which includes blocking using BLOXALL Blocking solution and normal horse serum (2.5%) as per manufacturer specifications. The two primary antibodies used were a mouse anti-human albumin antibody (ThermoFisher, MA1-19174) and a rabbit anti-human Ku80 antibody (Cell Signaling Technology, 2753S), diluted at 1:200 and 1:100, respectively, in MAXBind-Staining solution (Active Motif, 15253) and incubated overnight at 4 °C, as per manufacturer specifications. Secondary antibodies used were an anti-mouse IgG antibody conjugated to AP, and an anti-rabbit IgG antibody conjugated to HRP (Vector Laboratories, MP-7714), as a ready-to-use solution, incubated 10 min at RT. After washing, ImmPACT DAB substrate was added and allowed to develop for 2 min at RT. Slides were washed, ImmPACT Vector Red substrate was added and allowed to develop for 4 min, at RT. Slides were rinsed with water, and a hematoxylin counterstain was then applied.

Positive tissue controls (human tissues) negative tissue controls (non-IUTx sheep tissues) were stained and evaluated in parallel with every experiment performed evaluating experimental tissue slides. Details regarding antibodies used, dilutions, and methods for validation appear in Supplementary Table 7. Tissues sections were imaged using a LEICA DM4000B brightfield microscope (20X magnification) and positive cells identified/quantified by Image J.

## Antigen-specific $T_{h1}$ and $T_{h2}$ cell ELISpot

Reactive memory $T_{h1}$ and $T_{h2}$ responses to ET3 were determined by IFN-γ (MabTech, 3119-4APW-2) and IL-4 (MabTech, 3118-2 A) ELISpot assays as previously described[53,54]. PBMC were isolated from whole blood using Histopaque 1077 (Sigma Aldrich, 1077), following manufacturer's guidelines. PBMCs were then cultured in one of three conditions: without a stimulating antigen (No Stimulus control), with ET3, or in the presence of the non-specific mitogen phytohemagglutinin-L (PHA-L) (Sigma Aldrich, C11249738001) to test for cell anergy. Working concentrations of ET3 and PHA-L were 10 μg/mL. After completing the 40 h co-culture, the plate was washed, and anti-ovine IFN-γ and IL-4 added, followed by secondary antibody conjugated to ALP, and spots visualized via enzymatic cleavage of ALP substrate. A positive response was defined when both of the following criteria were met: (1) the ratio between the number of spot-forming units (SFU) in the presence and absence of ET3 is > 2 (Stimulation Index >2); and (2) a threshold minimum of 10 SFU/2 × 10^5 PBMC was reached.

## One-way mixed lymphocyte reaction

To determine whether IUTx recipients were tolerant to donor PLC, we tested proliferative responses via Mixed Lymphocyte Reaction (MLR) using a 5-bromo-2'-deoxyuridine (BrdU) ELISA (Roche,12352200). An MLR provides an efficient and highly specific in vitro model for the study of T cell proliferation in response to either allogeneic or xenogeneic antigen. For these assays, the *Responder* cells were PBMC isolated from IUTx recipients and non-IUTx control animals. *Stimulator* cells were: (1) same-donor non-transduced PLC; (2) same-donor PLC transduced with LV vector encoding mcoET3; (3) xenogeneic (third-party human) PBMC, and (4) PHA-L as a stimulating mitogen. Responder cells were also cultured alone to determine baseline levels of BrdU incorporation over the incubation period. After seeding 96-well plates with Stimulator cell types 1 through 3 in triplicate, cell cycle arrest (*in the Stimulator cells only*) was induced by adding mitomycin C (Stem Cell Technologies, 73274). The concentration of mitomycin C necessary to specifically halt cell division in PLC was empirically derived, testing mitomycin C concentrations ranging from 0.5 to 50 ng/mL. After thorough washing of stimulator cells, responder PBMC were added and co-cultured at a 10:1 (Responder:Stimulator) ratio. MLR co-culture proceeded for 120 h in IMDM 10% FBS, then cells

were labeled overnight with the thymidine analog BrdU, and tested for differential absorbance via ELISA as per manufacturer's guidelines. The stimulation index (SI) was calculated based on typical SI criteria for MLR cultures where: SI = (Stimulator + Responder) / (Responder).

## Luminex ID assay for detection of anti-HLA class I and class II antibodies

To determine whether IUTx recipients developed an HLA-directed antibody response due to exposure to human PLC, a Luminex assay for detection of HLA Class I and Class II IgG antibodies was performed according to manufacturer specifications (Immucor, 628200). Briefly, 1.2 μm multiscreen filter plates (Millipore Sigma, MSBVN1210) were used to incubate HLA bead mix provided by the manufacturer together with serum from IUTx recipients at the designated ratio. After 30 min incubation, plate wells were washed three times (wash buffer: phosphate-based buffer with NaCl, Tween-20, NaN$_3$, and BSA). Then, a sheep IgG-phycoerythrin antibody (R&D systems, F0126) conjugate was added in wash buffer for an additional incubation for 30 min. Afterwards, the prescribed volume of wash buffer was added, mixing to resuspend the beads, sample fluorescence was determined with a Luminex 200 Fluoroanalyzer, and analysis performed using the Luminex antibody software. The mean fluorescence intensity (MFI) value of each bead was compared to three cut-off values to establish background adjustment factors (BAFs). The cut-off values were calculated from the determined background of three negative control beads in each test well. This was repeated for each of the two beads to obtain three results or Adjusted MFI (AMFI) values. A sample was considered positive if two or more AMFI values were positive. A sample was considered negative if two or more AMFI values were negative.

## Tissue histopathology

A specialized veterinary pathologist performed gross anatomical examination to determine the presence of therapy-related morphologic abnormalities and evaluated hematoxylin and eosin (H&E)-stained slides of liver, lungs, spleen, thymus, intestine, and brain to identify potential histopathologic alterations.

## Generation of hemophilia A animals

We have previously described the establishment of a line of HA sheep harboring a premature stop codon and frame-shift mutation in exon 14 of the FVIII gene. Clinically these animals closely mimic severe human HA, consistently exhibiting a severe bleeding phenotype, with spontaneous hemarthrosis, leading to reduced locomotion, muscular hematomas, and episodes of hematuria. In addition, if these HA sheep are not treated shortly after birth, the animals will die from internal or umbilical cord bleeding. To generate HA animals, HA carriers were bred through natural breeding, or through cloning using somatic cell nuclear transfer.

## Statistics and reproducibility

All data shown, unless otherwise specified, are presented as mean +/− standard error of the mean (SEM). GraphPad PRISM v8 software was used to perform all statistical analyses. Comparisons between experimental results were determined either by two-tailed Student's $t$-test, for single comparisons, or by one-way ANOVA followed by post hoc analysis using Tukey's method for data involving multiple comparisons, to determine individual $p$ values. $p$ value $< 0.05$ was considered statistically significant.

Detection of specific cell markers and quantification of Human Ku80 (nuclear antigen)-positive cells in liver of IUTx animals by immunohistochemistry using chromogenic detection were performed in slides from all animals, and from positive (human) and negative (non-treated sheep) controls in at least three different experiments. In each slide quantification was performed in three regions of interest (ROI). To eliminate observer bias, an ImageJ script was utilized to process the ROI (ROIs) and identify positive cells. The script deconvoluted each ROI image into an H&E image and a DAB image, then considered positive cells using size, circularity, and color thresholding. In detail, specific macros with multiple functions were created to automate a series of tasks that included normalization, processing, and analysis of a large number of images. The normalizing step used techniques such as color deconvolution, which separates image channels to up to 3 specific colors. Tasks such as sharpening and contrast leveling to gain intuition for downstream analysis were also used. A "subtract background" function to isolate and analyze features in an image above a uniform background intensity was also applied, as well as the "after" function with auto thresholding algorithms, to segment the image into different regions based on pixel intensity values. The "watershed" function was used to separate touching or overlapping objects in the image. After normalizing each image, the "analyze particles" function was used to identify and measure features in the image, with options for size range, shape, and identity chosen based on visualization of the positive signal. In some cases, it was necessary to further fine-tune the analysis parameters to obtain accurate results, which was done by "training" each dataset. The macro was run multiple times, adjusting the parameters as needed to improve its overall performance. Unfortunately, no ground truth data was available, so parameters were manually adjusted after visual inspection of the results of the macro to ensure accuracy.

Detection of Human Ku80 (nuclear antigen)- and FVIII- positive cells by immunohistochemistry using immunofluorescence were performed in IUTx animals, and positive (human) and negative (non-treated sheep) controls in at least three different experiments. Quantification was performed using three different slides and counting a total of 1000–1500 cells.

For histopathological analysis (Fig. 8), whenever the amount of tissue harvested allowed, 2–4 tissue sections were sent per organ and analyzed by the pathologist.

## Reporting summary

Further information on research design is available in the Nature Portfolio Reporting Summary linked to this article.

## Data availability

The main data supporting the results of this study are available within the paper and in the Supplementary Information and Supplementary Data Files. Source data are provided with this paper. Next generation sequencing data generated in this study have been deposited into GEO under accession number GSE230081. https://www.ncbi.nlm.nih.gov/geo/query/acc.cgi Source data are provided with this paper.

## Code availability

The custom scripts for quantification of positive cells using ImageJ are available at GitHub: https://github.com/jhd8593/Martin_IHC.git

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

## Acknowledgements

This work was supported by NIH, NHLBI, HL135853, HL148681 M.R. and G.D.A. are the recipients of a fellow/mentor HHMI Gilliam Graduate Fellowships grant. B.T. is supported by a T32 pre-doctoral fellow position NIH NIBIB 2T32EB014836-06A. The Mick Hitchcock Ph.D. Nevada Proteomics Center is supported by NIH, NIGMS, GM103440. We would like to thank Wake Forest Baptist Health Special Hematology Laboratory for their excellent technical support and the performance of FVIII testing, the Wake Forest Institute for Regenerative Medicine Translational Core Manufacturing Center for providing the PLC used in these studies, Wake Forest ARP for the clinical care of the animals, and Rebecca Woolsey at The Mick Hitchcock Ph.D. Nevada Proteomics Center. We would like to thank Dr Samuel Rulli, and Dr. Martin Kreutz from Qiagen for their help with the cell differentiation evaluation experimental design, for performing RNA-seq library construction, NGS, and data Analysis.

## Author contributions

M.R., B.T., R.R., and S.K.G. executed experiments, data analysis and interpretation; J.D., D.M., D.R.Q., A.F., M.G., J.F., and D.S. provided technical expertise; J.A. provided reagents; C.B.D., H.T.S., and A.A. provided reagents and experimental feedback; M.R. drafted manuscript; G.A.P. and C.D.P. conception and experimental design, supervised experiments, performed data analysis and interpretation, wrote final version of the manuscript, and secured funding.

## Competing interests

C.B.D. and H.T.S. are co-founders of Expression Therapeutics, Inc. and own equity in the company. Dr. Denning is an employee of Expression Therapeutics, Inc. Expression Therapeutics licensed the intellectual property associated with the codon optimized ET3 transgene and HCB promoter. The terms of this arrangement have been reviewed and approved by Emory University in accordance with its conflict-of-interest policies. The rest of the co-authors have no competing financial and non-financial interests.
