## [Peer review file · Nature Communications]

REVIEWER COMMENTS

Reviewer #1 (Remarks to the Author):

This paper by Rodriguez et al represents a substantial advance for in utero cell therapy. In the large animal study, human placental cells engineered to express a highly secret variant of coagulation factor VIII were transplanted in utero and provided sustained therapeutic levels of factor VIII activity after birth. The study further supports safety of the approach. The fact that the human cells were not rejected in an all model is highly encouraging.

Some suggestions for further improvement of the manuscript:

1. The animals that were followed beyond birth were all he-statically normal. That should be mentioned earlier in the text to avoid confusion, as the intro is heavy on explanations about hemophilia A.
2. The authors show that the transplanted cells can be found in multiple organs. What is the phenotype of these cells, and does that differ depending on the organ/microenvironment? Do they represent or resemble a specific cell type?
3. It should be clearly stated that confirmation of some of the results in a disease model is desirable to reach definitive conclusions about immune responses to factor VIII and phenotypic correction.

Reviewer #2 (Remarks to the Author):

Hemophilia A (HA) is the most common X-linked bleeding disorder and development of an effective treatment before birth is an outstanding and worthy idea. In this study, authors transplanted human placental derived mesenchymal stromal cells expressing a bioengineered FVIII (PLC-mcoET3) into fetal sheep, and characterized the expression of plasma FVIII levels after birth. They reported that PLC-mcoET3 engrafted in major organs and the recipients did not mount a humoral or cellular response, However, there are several major flaws in this study that significantly compromised the conclusions made by the authors. Primarily, the study appears to have been conducted in wild type sheep, not in a disease animal model,(although there is an unclear reference to HA bred sheep) therefore the “% increase in FVIII over control” is a misleading measure over disease phenotype correction. It was unclear

if aPTT change in the wild type animals is correlated to disease correction in disease animals. Moreover, the tolerance studies are not well designed and the conclusions based on the current study are not accurate.

Major points:

1. "Results in Sustained Curative fVIII Plasma Levels" in the Title is an overstatement and misleading since the study was not done in a disease model. The entire study should be conducted in the HA sheep model to make conclusions such as "Sustained Curative fVIII Plasma Levels".
2. Critical technical details of PLC are missing. How were PLC isolated and cultured? Characterization of PLC? CD117 expression and other MSC markers should be presented. What is the transduction efficiency of PLC after being transduced with a lentiviral vector encoding mcoET3? What is the copy number of FVIII vectors in PLC? The transduction efficiency is a critical parameter of the product quality control and the FVIII expression level should be carefully characterized and presented for this study.
3. In Figure 1, Plasma FVIII activity in transplanted and normal control sheep was determined by activated Partial Thromboplastin Time (aPTT) one-stage assay. The authors should make it clear that aPTT is not equivalent to plasma FVIII activity or plasma FVIII levels in wild type sheep.

Figure 1a, c, d Y axis labeled as "% increase in FVIII over control" is not accurate. How exactly this value was calculated should be presented. The current Figure 1 Y axis "% increase in FVIII over control" indicates the absolute FVIII amount or activity which should be measure by ELISA, western blot, LC-MS or FVIII activity assay. aPTT is only an indirect measure of the overall coagulation function of all coagulation factors. Making the blood of normal sheep "clot 5% faster" is not equivalent to "5% of FVIII circulation" that changes the disease phenotype as the authors stated. Therefore, the dashed line of aPTT increase (5% increase in FVIII over control) in figures 1a b d is misleading "to be curative". Absolute value of aPTT should be presented.
4. Figure 1e showed the presence of ET3 protein in the plasma of treated animals by LC-MS, with the Y axis being "Normalized exclusive intensity". What does this value mean? The authors should measure native sheep FVIII using the same LC-MS method and present what is the percentage of ET3 in total blood FVIII.
5. One of the most critical studies in this paper is to analyze whether the presence of ET3 protein in circulation would induce anti-ET3 IgM and IgG antibodies. However, since the entire study was done in wild type sheep, extensive and appropriate controls should be included. Is it possible that because the engineered ET3 FVIII and native sheep FVIII are similar enough ET3 FVIII would not induce detectable immune response? The authors should challenge wild type adult sheep with ET3 FVIII isolated from engineered PLC-mcoET3 culture and conduct the tolerance assays.
6. In Figure 5, the location of high magnification of IHC images should be labeled on a lower magnification image. HE staining of and adjacent slide should be included to show the location of the PLC engraftment in the organs. ET3 IHC higher quality images should be provided. The phenotype of these engrafted human cells should be carefully characterized, as in general mesenchymal stromal cells

from all sources are not thought to engraft . Figure 5g, in addition to showing the isotype control, non-IUT treated animals and wild type sheep should be provided as the controls. In Figure 5g-h was anti-human FVIII staining. Please provide the antibody clone number. IHC results with anti-ET3 antibody should be provided as well so that human ET3 and sheep FVIII should be distinguished to highlight the expression of the transduced protein.

7. only IHC and PLC engraftment at birth was shown. 1 year, 2 year and 3 year IHC should be characterized and shown as well. Are the PLCs still engrafted after 3 years? Since the aPTT results seem to show the persistent high expression, PLC should be still engrafted after 3 years.

8. Experiments and explanations regarding the mechanisms through which PLC migrate to major organs are needed. The authors should characterize the phenotype of the transplanted PLCs in different organs

9. Engraftment rate: the authors used RT-qPCR using primers specific for mcoET3 and sheep GAPDH and by comparing ΔC_t values to a standard curve consisting of increasing percentages of mcoET3-PLC (0, 0.01, 0.1, 1, 5, and 10%) in sheep stromal cells. The authors should quantify the engraftment rate by human DNA amount in sheep tissues.

10. In Figure 5. Photos and descriptions of the morphology of the major organs from 18002 shall be provided to show if there is any sign of tumorigenesis, considering on average 25% of the liver and 35% of the lung (by RT-PCR) are transplanted cells.

Minor

1. The cell dosing was not clear. The authors said the cells were transplanted at the cell doses of 10^7 - 4×10^8 cell/kg in 500uL of QBSF-60. Was this dose based on the body weight of the pregnant sheep? Did all sheep receive the same volume of cell suspension with different cellular density according to the body weight? Details should be provided.

With the appropriate additional details this would be an important paper that would change the known experience with mesenchymal stromal cells.

Point-by-point response to the reviewer's comments

We thank both reviewers for their careful and thoughtful evaluation of our work, and for their criticisms and suggestions, that have allowed us to re-submit a much-improved version of the manuscript. As you can see below in the “detailed answers to the reviewers”, we have addressed all the questions that were raised, and we have modified the manuscript accordingly.

Reviewer #1 (Remarks to the Author):

This paper by Rodriguez et al. represents a substantial advance for in utero cell therapy. In the large animal study, human placental cells engineered to express a highly secret variant of coagulation factor VIII were transplanted in utero and provided sustained therapeutic levels of factor VIII activity after birth. The study further supports safety of the approach. The fact that the human cells were not rejected in an all model is highly encouraging.

Some suggestions for further improvement of the manuscript:

1. The animals that were followed beyond birth were all hemostatically normal. That should be mentioned earlier in the text to avoid confusion, as the intro is heavy on explanations about hemophilia A.

Thank you for this suggestion; we have now made this point very clear right from the introduction: “*using wild type sheep as large animal model of IUTx*”

2. The authors show that the transplanted cells can be found in multiple organs. What is the phenotype of these cells, and does that differ depending on the organ/microenvironment? Do they represent or resemble a specific cell type?

Thank you very much for this question. To address this important issue, we isolated RNA from the main tissues from the animals and created transcriptome RNA-seq libraries, which were aligned to *Homo sapiens* libraries after eliminating any possible cross-reactivity with ovine transcripts. We then compared the transcripts of the transduced PLC prior to transplant, with human transcripts known to be expressed in specific cells in each of the different organs. We found that, although several cell-specific transcripts were upregulated in the PLCs that engrafted within each tissue, a whole transcriptome profile consistent with that of a particular fully differentiated tissue-specific cell type was not present. We have now included these data as Supplemental Tables 2-6, which presents a comprehensive (human-specific) transcriptomic profile for Liver, Lung, Thymus, and Reproductive Tissue of these animals.

3. It should be clearly stated that confirmation of some of the results in a disease model

is desirable to reach definitive conclusions about immune responses to factor VIII and phenotypic correction.

We completely agree with this reviewer, and we have now added the following sentence to the end of the Discussion to underscore the importance of testing this therapy in HA animals:

“Nevertheless, confirmation of phenotypic correction by evaluating whether the annualized bleeding rate is significantly reduced from baseline throughout the lifetime of HA animals, and evidence that an immune response is still not present in these animals is a must to ascertain the true clinical impact of this prenatal approach.”

Reviewer #2 (Remarks to the Author):

Hemophilia A (HA) is the most common X-linked bleeding disorder and development of an effective treatment before birth is an outstanding and worthy idea. In this study, authors transplanted human placental derived mesenchymal stromal cells expressing a bioengineered FVIII (PLC-mcoET3) into fetal sheep and characterized the expression of plasma FVIII levels after birth. They reported that PLC-mcoET3 engrafted in major organs and the recipients did not mount a humoral or cellular response, However, there are several major flaws in this study that significantly compromised the conclusions made by the authors. Primarily, the study appears to have been conducted in wild type sheep, not in a disease animal model (although there is an unclear reference to HA bred sheep); therefore, the “% increase in FVIII over control” is a misleading measure over disease phenotype correction. It was unclear if aPTT change in the wild type animals is correlated to disease correction in disease animals. Moreover, the tolerance studies are not well designed, and the conclusions based on the current study are not accurate.

Major points:

1. “Results in Sustained Curative fVIII Plasma Levels” in the Title is an overstatement and misleading since the study was not done in a disease model. The entire study should be conducted in the HA sheep model to make conclusions such as “Sustained Curative fVIII Plasma Levels”.

We apologize if the title or any text in the manuscript were misleading. That was certainly not our intention, as we had stated that “*evaluation of plasma FVIII activity at intervals throughout the duration of the study, was expressed as % increase over non-transplanted control*”. But we completely understand and agree with the point that this reviewer makes, and we have now changed the title to read “...*Results in Increased fVIII Plasma Levels...*” We also removed wording such as “curative” or “therapeutic” from the main text and figures when referring to the results obtained in these studies.

2. Critical technical details of PLC are missing. How were PLC isolated and cultured? Characterization of PLC? CD117 expression and other MSC markers should be presented. What is the transduction efficiency of PLC after being transduced with a lentiviral vector encoding mcoET3? What is the copy number of FVIII vectors in PLC? The transduction efficiency is a critical parameter of the product quality control and the FVIII expression level should be carefully characterized and presented for this study.

We apologize we were not clear regarding this very important point. We used one master cell bank, the isolation, culture, characterization, and transduction of which with lentiviral vectors encoding mcoET3 was described in detail in PMID: 32258210. We have now made this point clear in the main text of the manuscript.

Thank you for bringing the important point regarding quality control of the transduced product to our attention. Because the lentiviral vector does not encode any reporter gene, transduction efficiency cannot be extrapolated. Instead, we used the vector copy number (VCN) <5 and production of FVIII / 10^6 cells/24 hours > 5 IU as release criteria for this study.

In Supplementary Table 1, please find the specific VCN and IU of FVIII/ 10^6 cells/24 hours calculated prior to each injection.

We have also added this information to the Materials and Methods.

3. In Figure 1, Plasma FVIII activity in transplanted and normal control sheep was determined by activated Partial Thromboplastin Time (aPTT) one-stage assay. The authors should make it clear that aPTT is not equivalent to plasma FVIII activity or plasma FVIII levels in wild type sheep.

Figure 1a, c, d Y axis labeled as “% increase in FVIII over control” is not accurate. How exactly this value was calculated should be presented. The current Figure 1 Y axis “% increase in FVIII over control” indicates the absolute FVIII amount or activity which should be measure by ELISA, western blot, LC-MS or FVIII activity assay. aPTT is only an indirect measure of the overall coagulation function of all coagulation factors.

Making the blood of normal sheep “clot 5% faster” is not equivalent to “5% of FVIII circulation” that changes the disease phenotype as the authors stated. Therefore, the dashed line of aPTT increase (5% increase in FVIII over control) in figures 1a b d is misleading “to be curative”. Absolute value of aPTT should be presented.

We thank the reviewer for bringing these points to our attention.

All FVIII levels were determined by performing FVIII activity assays, and we have better clarified this information in the revised manuscript.

We have now included the formula for how the percentage increase of FVIII activity was calculated in the Materials and Methods under “Factor VIII (FVIII) Activity Assay”, and we also made sure that it is clear that what is being represented in the Fig is “%

increase of FVIII activity”, by renaming the axis “% increase in FVIII activity over control”.

The 5% line denoted 5% increase in FVIII activity, and not 5% decrease in overall clotting time. We apologize if this was not clear. Nevertheless, we have removed the 5% lines from all 3 graphics.

As mentioned above, aPTTs to measure whole coagulation time were not performed at any time point, as it would not be relevant, as stated by this reviewer. The % increase in FVIII activity are displayed in Figures 1a, c, and d.

4. Figure 1e showed the presence of ET3 protein in the plasma of treated animals by LC-MS, with the Y axis being “Normalized exclusive intensity”. What does this value mean? The authors should measure native sheep FVIII using the same LC-MS method and present what is the percentage of ET3 in total blood FVIII.

We used LC-MS as a targeted approach to confirm/disprove the presence of the ET3 protein in plasma samples. The Scaffold DIA normalization algorithm for intensity quantification was applied to appropriately adjust values for comparison purposes. Summed intensities from different MS samples are first log₁₀ transformed, and a histogram is then built for each sample. Quarterlies for each sample are calculated and compared to the quarterlies of the entire experiment. The exclusive intensity is the summarized intensity value of only the peptides that are associated with this protein. We have now included this information in the Results section, Figure Legend, and in the Materials & Methods section.

5. One of the most critical studies in this paper is to analyze whether the presence of ET3 protein in circulation would induce anti-ET3 IgM and IgG antibodies. However, since the entire study was done in wild type sheep, extensive and appropriate controls should be included. Is it possible that because the engineered ET3 FVIII and native sheep FVIII are similar enough ET3 FVIII would not induce detectable immune response? The authors should challenge wild type adult sheep with ET3 FVIII isolated from engineered PLC-mcoET3 culture and conduct the tolerance assays.

This is an extremely relevant point, and we thank this Reviewer from bringing it up. We have previously performed studies proving that administration of either ET3 or hFVIII protein to wild type juvenile sheep induces high titer IgGs starting at week 2 of infusion, and that these IgGs are inhibitory antibodies to ET3 or FVIII. In addition, we showed that the PLC-mcoET3 that were used in the prenatal studies, when injected IP into wildtype juvenile sheep, also induced the generation of specific ET3/FVIII IgGs in 66% of the animals by week 2 after administration.

Since these studies involve large animals and the IACUC at our Institution strongly advises against duplication of the same overall studies whenever possible, and the prior data had clearly demonstrated that wildtype sheep developed an immune response to the protein when administered by bolus, and through the same exact cells, we did not repeat those experiments in these studies.

In order to make this point clear, we now added the following information to the Results section: *“Administration of ET3 or hFVIII protein to wildtype juvenile sheep induces a strong immune anti-ET3/FVIII IgG response starting at week 2 post infusion and leads to the development of inhibitory antibodies¹³”*. We hope that it is now clear to the readers that both ET3 and hFVIII protein are highly immunogenic when administered to wildtype juvenile sheep.

In addition, we also added the following sentence at the end of the Discussion: *“Nevertheless, confirmation of phenotypic correction by evaluating whether the annualized bleeding rate is significantly reduced from baseline throughout the lifetime of HA animals, and evidence that an immune response is still not present in these animals is a must to ascertain the true clinical impact of this prenatal approach”*.

6.1 In Figure 5, the location of high magnification of IHC images should be labeled on a lower magnification image. HE staining of and adjacent slide should be included to show the location of the PLC engraftment in the organs.

We have now provided a new Figure 6, which depicts both low and higher magnification images of liver tissue sections stained with an antibody specific to the human nuclear antigen Ku80, using DAB as a chromogenic dye and hematoxylin counterstain, which allows the simultaneous visualization of the positive (human) nuclei and the localization of these positive nuclei within the liver tissue.

6.2 ET3 IHC higher quality images should be provided.

We are unsure why the quality of this image seen by this reviewer is poor, as we uploaded a very high-resolution image, but we thank the reviewer for letting us know. We have uploaded a new high-resolution image.

6.3 The phenotype of these engrafted human cells should be carefully characterized, as in general mesenchymal stromal cells from all sources are not thought to engraft. Please see the studies performed to answer this question under question 8.2.

6.3 In Figure 5g-h was anti-human FVIII staining. Please provide the antibody clone number.

We apologize for only having provided this information in the Supplementary Materials and Methods. We have now added this information to the manuscript text under Materials and Methods.

6.4 IHC results with anti-ET3 antibody should be provided as well so that human ET3 and sheep FVIII should be distinguished to highlight the expression of the transduced protein. Figure 5g, in addition to showing the isotype control, non-IUTx treated animals and wildtype sheep should be provided as the controls.

The reviewer makes a very important point, which we did not make clear. Overall, there is only a 9% sequence modification between ET3 protein and hFVIII. These modifications were made through substitutions residing in the hFVIII A1 and ap-A3

domains and in the SQ linker sequence for the B-domain. The antibody used targeted the FVIII C2 domain, the sequence of which is identical between human FVIII and ET3. Therefore, the antibody used targets both human FVIII and ET3. The images displayed in Fig 5g-h are the outcome of anti-FVIII/ET3 staining of a tissue section from the liver of the hemophilia A animal that was transplanted in utero. As previously reported, this line of animals is cross-reactive material (CRM)-negative, meaning that FVIII protein is completely absent from circulation and from their tissues; as such, any FVIII staining is derived from the presence of PLC-mcoET3.

Thus, to detect the presence of human cells in the HA animal we could perform FVIII/ET3 staining, while in wildtype animals, we had to use an antibody to the human nuclear antigen Ku80 to specifically detect the engrafted cells. That is also the reason why Fig 5g is an isotype control with tissue derived from the same animal and not a non-transplanted wildtype animal.

We have now provided Supplementary Figure 4, which shows staining of 2 non-transplanted hemophilia A (HA) animals and a wildtype animal, demonstrating that the mutation in HA sheep results in complete lack of FVIII protein (CRM-negative).

The following text has also been added to the manuscript to clarify these points:

“The production and presence of ET3 protein in liver was also determined in the HA animal (Fig. 5g,h), as animals with this mutation lack factor VIII antigen (cross-reactive material negative) (Supplementary Figure 4), thereby ensuring that detection of FVIII/ET3 in this animal is the result of production of ET3/FVIII by the transplanted cells present in the organ.”

7. only IHC and PLC engraftment at birth was shown. 1 year, 2 year, and 3-year IHC should be characterized and shown as well. Are the PLCs still engrafted after 3 years? Since the aPTT results seem to show the persistent high expression, PLC should be still engrafted after 3 years.

We did not have these data at the time of the initial submission of this manuscript, because we were collecting data on the levels of FVIII activity in these animals for as long we could prior to euthanizing the animals. In order to address this important question raised by this Reviewer, we have now euthanized animals between 2.33- and 5-years post-IUTx and show that the PLC are still engrafted in the tissues at these time points (Supplementary Table 1).

We have now included these data in the Results section and have shown that the PLC-mcoET3 are present in these animals (Fig. 7 and Supplementary Fig. 6).

8.1. Experiments and explanations regarding the mechanisms through which PLC migrate to major organs are needed.

We thank the reviewer for this comment. These studies did not aim to investigate

mechanisms of absorption/transport of cells in and out of the peritoneal cavity, as this route of administration is well documented, it is clinically approved, and in the past has been used successfully in the transplantation of human fetuses (for examples, please see PMID: 8943162, PMID: 8942778, PMID: 1504665). Nevertheless, a brief explanation of the rationale for using this route of administration follows. The peritoneal cavity is surrounded by 2 serous membranes, the parietal peritoneum, and the visceral peritoneum. These membranes consist of a layer of mesothelial cells on a connective tissue base which is perfused with circulatory and lymphatic vessels. The parietal peritoneum gets its blood supply from lumbar, intercostals, and epigastric regions of the abdominal wall and drains into the inferior vena cava, while the visceral peritoneum receives its blood supply from the superior mesenteric artery and drains into the portal vein. Constitutive migration of B cells from the circulation into the peritoneal cavity and back has previously been reported by these and other investigators (PMID: 18250426, PMID: 17289810). Thus, the traffic of cells from the peritoneal cavity to the blood supply is well documented.

The lymphatic system is also known to serve as a route for migration of cells in and out of the peritoneal cavity. The subdiaphragmatic lymphatic system is responsible for 70-80% of the lymphatic flow from the peritoneal cavity. The lymphatic stomata (small openings of lymphatic capillaries on the free surface of the mesothelium) exist in the peritoneum, and cells that are present or that are administered into the peritoneal cavity can circulate through the lymphatic vessels into the thoracic duct which drain into the subclavian vein. In addition, cells present and/or administered to the peritoneal cavity can also circulate through the intestinal lymphatic capillaries and the associated collecting vessels in the mesentery. The mesentery vessels also drain to the thoracic duct. Thus, cells that enter the intestinal lymphatic capillaries also reach the thoracic duct, and subsequently enter the main blood circulation.

Supplementary Fig. 6 shows the percentage of cells engrafted in thoracic and mesenteric lymph nodes using the same methodology described for the other tissues

Of note is that we and others have shown in prior studies using the fetal sheep model how the route of administration and/or adhesion molecules impact the sites and level of engraftment of different cells following IUTx (PMID: 17705296, PMID: 22488442, PMID: 27304918, PMID: 23413357, PMID: 19383418).

8.2 and 6.3. The phenotype of these engrafted human cells should be carefully characterized, as in general mesenchymal stromal cells from all sources are not thought to engraft. The authors should characterize the phenotype of the transplanted PLCs in different organs.

Thank you for this important question. The goal of this study is to test whether administration of PLC-mcoET3 to fetal recipients is safe and can provide long-lasting elevated levels of ET3/FVIII protein in circulation, as detected by an increase in FVIII activity in plasma. Since the mcoET3 transgene is not under a tissue-specific promoter, even if PLC-mcoET3 switch phenotypes into tissue-specific cells, this will not shut down the transcription of the transgene and production of the ET/FVIII protein. Nevertheless,

we agree with the Reviewer that this is a very important point. We now provide data from 2 types of studies to address this Reviewer's question. In the first study, RNA was isolated from the main tissues from IUTx animals, and transcriptome RNA-seq libraries were created, which were aligned to *Homo sapiens* libraries after eliminating any possible cross-reactivity with ovine transcripts. We then compared the transcripts of the transduced PLC prior to transplant and the human transcripts found in the different IUTx animal tissues with human transcripts known to be expressed in specific cells within each of the different organs. We found that, although several cell-specific transcripts were upregulated in the PLCs that engrafted within each tissue, a whole transcriptome profile consistent with that of a particular fully differentiated tissue-specific cell type was not present. We have now included these data as Supplemental Tables 2-6, which presents a comprehensive (human-specific) transcriptomic profile for Liver, Lung, Thymus, and Reproductive Tissue of these animals.

We also performed IHC with antibodies specific for some of transcripts found to be present in hepatocytes, such as albumin and Urea cycle enzyme CPS1, and for LYVE-1, which is known to be a marker of liver sinusoidal endothelial cells.

Cells positive for LYVE-1 and for Urea cycle enzyme CPS1 were found within the liver sections of the transplanted animals but staining for human albumin was unable to detect human hepatocytes producing human albumin in animals that had RNA transcripts for this protein.

Since the liver in sheep weighs approximately 1Kg, it would be impossible to absolutely exclude the possibility of the existence of fully differentiated cells. Nevertheless, it is important to note that these same regions contained cells positive for the human nuclear antigen Ku80. Collectively, the data show that transplanted PLC-mcoET3 express some transcripts and produce some of the proteins of cells specific to the tissue in which they lodge, but the data cannot confirm the presence of a fully functional differentiated organ-specific cell.

9. Engraftment rate: the authors used RT-qPCR using primers specific for mcoET3 and sheep GAPDH and by comparing ΔC_t values to a standard curve consisting of increasing percentages of mcoET3-PLC (0, 0.01, 0.1, 1, 5, and 10%) in sheep stromal cells. The authors should quantify the engraftment rate by human DNA amount in sheep tissues.

As requested, we have now included data (Supplementary Fig. 2 and Fig. 7) with the engraftment levels in sheep tissues measured by qPCR to quantify human DNA.

10. In Figure 5. Photos and descriptions of the morphology of the major organs from 18002 shall be provided to show if there is any sign of tumorigenesis, considering on average 25% of the liver and 35% of the lung (by RT-PCR) are transplanted cells.

We requested photos from the necropsy of animal 18002 from the Department of Pathology - Section on Comparative Medicine - Animal Resources Program, who

performs the necropsies independently from the investigator. Investigators have access to the tissues only after the necropsy is completed by an independent pathologist. Unfortunately, photos depicting the gross morphology of the organs of this animal are not available, as we were told that pictures are only taken in the event that anomalies are found.

We have attached a copy of the pathology report at the end of this document. Histological analysis of tissues were performed by Animal Resources Program and by another independent animal pathologist, and no signs of tumorigenesis were found. Histological images of the tissues from this animal can be found in Figure 6.

Minor

1. The cell dosing was not clear. The authors said the cells were transplanted at the cell doses of 10^7 - 4×10^8 cell/kg in 500uL of QBSF-60. Was this dose based on the body weight of the pregnant sheep? Did all sheep receive the same volume of cell suspension with different cellular density according to the body weight? Details should be provided.

We apologize for not being clear. The fetus at the time of transplant weighs approximately 100g. Thus, all the animals received the same dose/kg calculated for 100g of body weight. We have now added a note to Supplemental Table 1 to clarify this question.

With the appropriate additional details this would be an important paper that would change the known experience with mesenchymal stromal cells. Thank you very much for your questions and suggestions.

Wake Forest University School of Medicine
Department of Pathology - Section on Comparative Medicine - Animal Resources Program

(This report is confidential-- if you received it in error, please shred immediately)

Final Necropsy Report

Necropsy # AR18-2005

Necropsy #	AR18-2005	Campus/Building/Pen or Room	DC/A1/G52
Date of Death	02/21/2018	Investigator	Christopher D. Porada
Date of Necropsy	02/21/2018	Department	WFIRM
Species	Ovis aries (Domestic sheep)	Experiment#	
Strain/Breed	Merino	ACUC#	A17-146
Animal	18002	Clinician	ERIN MITCHELL
Origin		Prosector	CARSON SAKAMOTO
Sex	FEMALE	Primary Pathologist	CARSON SAKAMOTO
Age	0 Day(s)	Faculty Pathologist	NANCY KOCK, DVM, PhD
Arrival Date			

Clinical Findings:

This animal was the second of twin lambs delivered by cesarean section at full term. It was taken to the recovery room, dried off, had mucus removed from the nose and mouth, and had an oxygen mask placed while being stimulated. It was given Dopram and was intubated after it was observed only occasionally breathing on its own. Pupillary light reflexes were absent during this time. An intravenous catheter was placed and furosemide, Dopram, and dexamethasone were administered. CPR was attempted when its heart stopped, but was unsuccessful.

Experimental History Comments:

This animal was part of a protocol entitled, "Stem cell-based therapies to correct hemophilia A before birth."

Gross Findings:

Examined is the carcass of a 2.5kg female lamb with mild postmortem decomposition. The left lateral saphenous vein has an intravenous catheter with an extension set attached to a 3mL syringe. The lungs are diffusely red and sink in formalin. The tracheal mucosa is covered by a thin film of clear fluid. The gastrointestinal tract is empty except for meconium in the cecum and colon. The urinary bladder contains approximately 20mL of urine.

Laboratory Tests:

Bacteriology: Not done
Virology: Not done
Serology: Not done
Urinalysis: Not done
Hematology: Not done
Other Samples: Not done

Laboratory Results:

Gross Morphological Diagnoses:

Atelectasis, diffuse, marked, lung

Gross Interpretation / Comments:

The cause of inability to thrive is uncertain, but death was due to hypoxia due to failure to breathe following cesarean section. Histopathology is pending and additional comments will follow.

Histological Findings:

Slide #	Tissue	Slide Description
1	HEART	NSL
2	HEART	NSL
3	LUNG	Alveoli are closely apposed throughout and a few contain squamous epithelial cells.
4	THYMUS	NSL
5	KIDNEY	NSL
6	URINARY BLADDER	NSL
7	UMBILICAL CORD	NSL
8	ADRENAL GLAND	NSL

Final Necropsy Report

Necropsy # AR18-2005

Slide #	Tissue	Slide Description
9	COLON	NSL
10	BRAIN	NSL
11	BRAIN	NSL
12	BRAIN	NSL
13	BRAIN	NSL
14	BRAIN	NSL
15	BRAIN	NSL

Histological Morphological Diagnoses:

Atelectasis, diffuse, severe, lung
Intra-alveolar squamous epithelial cells, multifocal, minimal, lung

Final Diagnoses:

Pulmonary atelectasis

Final Interpretation Comments:

Some of these tissues were inadvertently not well preserved, but diffuse pulmonary atelectasis was evident and caused death. Deep respiratory movements elicited by any stressor, including hypoxia can cause in utero aspiration of squamous epithelial cells from the amniotic fluid, accounting for their presence in the lung.

Completed By: CARSON SAKAMOTO

Completed Date: 03/21/2018

Approved By: NANCY KOCK, DVM, PhD
Diplomate, American College of Veterinary Pathologists

Approved Date: 03/22/2018

REVIEWERS' COMMENTS

Reviewer #1 (Remarks to the Author):

The authors have addressed my questions and also improved data reporting, explanations, and several technical aspects of the study. The study provides interesting observations and strong support for the concept of in utero gene therapy using a large animal model. Lack of immune rejection and of antibody formation against the human FVIII are very interesting aspects.

Reviewer #2 (Remarks to the Author):

The paper is substantially improved after addressing reviewer comments.

While it remains unclear if the human mesenchymal stem cells actually engraft (a concept that goes against the prevailing data pertaining to the use of MSCs and even placental derived mscs) it does appear to demonstrate that they at least deliver the vector that generates an increase in Factor VIII production over historical wild type controls for a sustained period. It is a first step in the long path toward eventual human trials.

The provocative paper is worthy of publication.

Reviewer #3 (Remarks to the Author):

I was asked to perform a limited review of this manuscript to help address the question of whether the engineered human PLCs “engraft” in this lamb model of in utero transplantation. I suppose that depends on the definition of engraftment. What the authors have very convincingly demonstrated is that donor human PLCs persist and proliferate in various tissues in the recipient, do not induce specific immune response, and produce sustained levels of engineered ET3/FVIII for the duration of the study. They do not claim to demonstrate differentiation and histologic integration of the cells into various tissues leaving some question as to exactly what state the cells persist in and how they interact with whatever niche they reside in. They demonstrate dispersal of the cells between hepatocytes and within the endothelium of liver sinusoids among other locations, consistent with the mesenchymal origin of the cells. There is no evidence of clonal expansion or derangement of adjacent architecture in organs, and the cells seem to demonstrate very limited induction of tissue specific function arguing against any tissue specific differentiation.

I think the best interpretation is that they do engraft as tissue mesenchymal cells but maintain their overall phenotype and continue to perform their engineered function. What is extraordinary to me is the degree of proliferation that occurs over time (demonstrated both functionally and by human specific molecular and histologic techniques), and the specific tolerance demonstrated to these xenogeneic cells.

From a pragmatic perspective, does it really matter? If this is viewed as lodgment followed by expansion rather than engraftment, the ultimate potentially therapeutic goal would still be accomplished without apparent ill effect on any of the tissues they reside in.

In my view, this is a very significant contribution to the field of in utero transplantation, demonstrating the feasibility and potential efficacy of a strategy that has long been contemplated – the use of engineered mesenchymal cells to deliver a therapeutic product. In this instance the question of engraftment is of more scientific than practical importance and in my opinion the authors have done what is reasonable to clarify the fate of these cells.

NCOMMS-22-15370B now entitled “Transplanting FVIII/ET3-Secreting Cells in Fetal Sheep Increases FVIII Levels Long-Term Without Inducing Immunity or Toxicity”

Point-by-point response to the reviewer’s comments

We thank both reviewers for their careful and thoughtful evaluation of our work, and for their criticisms and suggestions, that have allowed us to re-submit a much-improved version of the manuscript. We are also pleased that the reviewers are now enthusiastic about the revised version of the manuscript.

Reviewer #1 (Remarks to the Author):

The authors have addressed my questions and also improved data reporting, explanations, and several technical aspects of the study. The study provides interesting observations and strong support for the concept of in utero gene therapy using a large animal model. Lack of immune rejection and of antibody formation against the human FVIII are very interesting aspects.

We thank reviewer 1 for all her/his previous suggestions that have allowed us to re-submit a much-improved version of the manuscript. We are also pleased by this reviewer’s comments that this study provides interesting observations and strong support for the concept of in utero gene therapy using a large animal model.

Reviewer #2 (Remarks to the Author):

The paper is substantially improved after addressing reviewer comments. While it remains unclear if the human mesenchymal stem cells actually engrafted (a concept that goes against the prevailing data pertaining to the use of MSCs and even placental derived mscs) it does appear to demonstrate that they at least deliver the vector that generates an increase in Factor VIII production over historical wild type controls for a sustained period. It is a first step in the long path toward eventual human trials. The provocative paper is worthy of publication.

We thank reviewer 2 for all her/his previous questions that have allowed us to re-submit a much-improved version of the manuscript. We are pleased by this reviewer’s support regarding the publication of the manuscript. We also understand this reviewer’s perspective in pointing out that many studies are unable to demonstrate that MSC can lodge/engraft in tissues for a long period of time. We believe that we have shown in an unbiased way that the engineered placental cells are present in the tissues. We have done this through many different methods, such as increase in FVIII activity, detection of ET3 protein in circulation, RT-PCR for the transgene qPCR, for the presence of human DNA in the tissues, we demonstrated the presence of human cell-specific transcripts through NGS and visualized the cells through IHC. Therefore, we are confident in our results. It is possible that these studies differ from those that report lack of MSC engraftment because here cells are given to animals that do not have an underlying inflammatory disease.

Reviewer #3 (Remarks to the Author):

I was asked to perform a limited review of this manuscript to help address the question of whether the engineered human PLCs “engraft” in this lamb model of in utero transplantation. I suppose that depends on the definition of engraftment.

What the authors have very convincingly demonstrated is that donor human PLCs persist and proliferate in various tissues in the recipient, do not induce specific immune response, and produce sustained levels of engineered ET3/FVIII for the duration of the study. They do not claim to demonstrate differentiation and histologic integration of the cells into various tissues leaving some question as to exactly what state the cells persist in and how they interact with whatever niche they reside in.

They demonstrate dispersal of the cells between hepatocytes and within the endothelium of liver sinusoids among other locations, consistent with the mesenchymal origin of the cells. There is no

evidence of clonal expansion or derangement of adjacent architecture in organs, and the cells seem to demonstrate very limited induction of tissue specific function arguing against any tissue specific differentiation.

I think the best interpretation is that they do engraft as tissue mesenchymal cells but maintain their overall phenotype and continue to perform their engineered function.

What is extraordinary to me is the degree of proliferation that occurs over time (demonstrated both functionally and by human specific molecular and histologic techniques), and the specific tolerance demonstrated to these xenogeneic cells.

From a pragmatic perspective does it really matter? If this is viewed as lodgment followed by expansion rather than engraftment, the ultimate potentially therapeutic goal would still be accomplished without apparent ill effect on any of the tissues they reside in.

In my view, this is a very significant contribution to the field of in utero transplantation, demonstrating the feasibility and potential efficacy of a strategy that has long been contemplated – the use of engineered mesenchymal cells to deliver a therapeutic product. In this instance the question of engraftment is of more scientific than practical importance and in my opinion the authors have done what is reasonable to clarify the fate of these cells.

We are grateful for the insightful comments provided by reviewer 3 and for his interpretation of the results presented. We are also very pleased that this reviewer thinks that the data presented in this manuscript are a very significant contribution to the field of in utero transplantation.